# Effects of a 3-Week Hospital-Controlled Very-Low-Calorie Diet in Severely Obese Patients

**DOI:** 10.3390/nu13124468

**Published:** 2021-12-14

**Authors:** Ivan Ožvald, Dragan Božičević, Lidija Duh, Ivana Vinković Vrček, Ivan Pavičić, Ana-Marija Domijan, Mirta Milić

**Affiliations:** 1Duga Resa Special Hospital for Extended Treatment, 47250 Duga Resa, Croatia; iozvald@gmail.com (I.O.); bozicevicdragan@gmail.com (D.B.); lidijaduh1@gmail.com (L.D.); 2Analytical Toxicology and Mineral Metabolism Unit, Institute for Medical Research and Occupational Health (IMROH), Ksaverska Cesta 2, 10001 Zagreb, Croatia; ivinkovic@imi.hr; 3Radiation Dosimetry and Radiobiology Unit, Institute for Medical Research and Occupational Health (IMROH), Ksaverska Cesta 2, 10001 Zagreb, Croatia; ipavicic@imi.hr; 4Department of Pharmaceutical Botany, Faculty of Pharmacy and Biochemistry, University of Zagreb, 10000 Zagreb, Croatia; adomijan@pharma.hr; 5Mutagenesis Unit, Institute for Medical Research and Occupational Health (IMROH), Ksaverska Cesta 2, 10001 Zagreb, Croatia

**Keywords:** cytochalasin B-blocked micronucleus *cytome* assay, DNA repair, DNA stability, excessive weight loss, low-calorie restriction diet

## Abstract

Although a very-low-calorie diet (VLCD) is considered safe and has demonstrated benefits among other types of diets, data are scarce concerning its effects on improving health and weight loss in severely obese patients. As part of the personalized weight loss program developed at the Duga Resa Special Hospital for Extended Treatment, Croatia, we evaluated anthropometric, biochemical, and permanent DNA damage parameters (assessed with the cytochalasin B-blocked micronucleus *cytome* assay—CBMN) in severely obese patients (BMI ≥ 35 kg m^−2^) after 3-weeks on a 567 kcal, hospital-controlled VLCD. This is the first study on the permanent genomic (in)stability in such VLCD patients. VLCDs caused significant decreases in weight (loss), parameters of the lipid profile, urea, insulin resistance, and reduced glutathione (GSH). Genomic instability parameters were lowered by half, reaching reference values usually found in the healthy population. A correlation was found between GSH decrease and reduced DNA damage. VLCDs revealed susceptible individuals with remaining higher DNA damage for further monitoring. In a highly heterogeneous group (class II and III in obesity, differences in weight, BMI, and other categories) consisting of 26 obese patients, the approach demonstrated its usefulness and benefits in health improvement, enabling an individual approach to further monitoring, diagnosis, treatment, and risk assessment based on changing anthropometric/biochemical VLCD parameters, and CBMN results.

## 1. Introduction

Non-communicable diseases (NCDs), characterized as non-transmissible with a long duration and slow progression, account for 63% of total deaths worldwide (up to 36 million of the 57 million total deaths according to the World Health Organization’s (WHO) Global Action Plan for the Prevention and Control of NCDs 2013–2020) [1]. In the development of the most common NCDs and other chronic diseases and comorbidities, a key role is played by inflammation, oxidative and DNA damage accumulation in cells and organs, while several major NCDs (diabetes mellitus type 2 (DMT2), coronary heart disease, stroke, asthma, and several cancers) have been demonstrated to have an association with obesity and an unhealthy diet [2,3,4,5,6,7]. Foods, diet, and nutritional status, including being overweight and obese, are also associated with the elevated parameters of lipid profile and insulin action resistance [8,9]. These conditions are not only risk factors for NCDs, but are themselves also major contributors of illness.

Obesity in Croatia is increasing rapidly, placing the country among those with the highest obesity prevalence in Europe [10,11,12,13]. With the lack of serious systemic monitoring of overweight/obese people, there is a need to develop a special program for monitoring the health, diet, and lifestyle habits of the Croatian (obese) population that would also generate a personalized motivation towards weight loss lifestyle changes. This need was recognized by the Duga Resa Special Hospital for Extended Treatment, Croatia, which developed a local program that is also open to accepting obese persons from other parts of Croatia and other regions. The program includes personalized 24 h medical surveillance while severely obese patients are performing a 3-week 567 kcal hospital-controlled very-low-calorie diet (VLCD). All patients’ meals are prepared fresh in the hospital kitchen, supervised by the hospital nutritionist. The program has been in place for five years and has proven to be a highly successful and low-cost program that improves patient wellbeing and offers a good opportunity to reduce the burdens on the healthcare system associated with incidences of severe chronic disease and comorbidities in obese people since a healthy diet is considered both a primary and secondary factor in the prevention of NCD (further) development.

This study aimed to evaluate the effects of the VLCD on markers of lipid profile, insulin action resistance, oxidative stress response by means of reactive oxygen species (ROS) and glutathione (GSH) levels, permanent DNA damage/stability, and anthropometric parameters in order to identify early biomarker(s) of patient status improvement during the 3-week VLCD. We were particularly interested in finding biomarker(s) to contribute towards developing a personalized and individualized VLCD approach.

Permanent DNA damage was measured in peripheral blood using the cytochalasin B-blocked micronucleus *cytome* (CBMN) assay. This assay is used for a wide range of developmental and degenerative diseases and has proven value as a predictor of the response, an adverse event, or the susceptibility to different stressors, demonstrating promise for widespread implementation into clinical practice [14,15,16]. The CBMN assay measures changes in apoptotic, necrotic, and mitotic index frequency and shows different types of DNA damage through the frequencies of the micronuclei (MNi)- (chromosome breaks/loss during cell division), nucleoplasmic bridges (NPBs)- (chromosome rearrangement markers), and nuclear buds (NBs)- (DNA amplification markers) [17,18]. As similar DNA damage and repair mechanisms are expected in different tissues [2,3], peripheral lymphocytes can serve as an excellent cell target due to their half-life and their presence in all body districts [19,20]. To the best of our knowledge, this is the first study involving human patients undergoing a 3-week 567 kcal hospital-controlled VLCD assessing genomic (in)stability through the CBMN assay together with anthropometric and biochemical parameters, and which compares these parameters before the start and on the last day of the diet period at both group and individual levels. The study revealed several possible biomarkers that could be used to track improvements of VLCDs, or even the patients that could be chosen for the program. The VLCD results can also facilitate the development of an individual approach, diagnosis, treatment, and risk assessment.

## 2. Materials and Methods

If not stated otherwise, all chemicals and materials were from Merck (former Sigma Aldrich, St. Louis, MI, USA).

### 2.1. Ethical Approval for the Study on Humans

The study was approved by the Ethics Committees of the Duga Resa Special Hospital for Extended Treatment (No.: 08-08-970/19), by the Institute for Medical Research and Occupational Health (IMROH, No.: 100-21/19-10), and by the School of Medicine, University of Zagreb, Croatia (No. 380-59-10106-20-111/174, class: 641-01/20-02/01). The latter approval is mandatory for all experiments involving human exposure at the national level. The study also holds Clinical Trials.gov number: NCT05055154 and study protocol number: OSMN35aVLCD-26, verified in September 2021.

### 2.2. Subject Recruitment and Diet

In total, 26 subjects were recruited in hospital according to the following inclusion criteria: individuals with BMI ≥ 35 kg m^−2^ accepted to stay in hospital for 3 weeks under constant medical surveillance, to practice a 567 kcal diet in hospital, to fill out questionnaires, and to give blood samples at the beginning and on the last day of the 3-week hospital stay. The study excluded pregnant women, minors, legally incapacitated persons, patients with currently tumorous diseases, or in a diagnostic procedure with ionizing radiation.

For the calorie intake, a standardized food questionnaire (EPIC-Norfolk) was used [21]. Participants answered questions on their average food intake within the past year. The questionnaire was adjusted to Croatian food and portions, with more than 130 FFQ items included (see Appendix A for the original FFQ questionnaire in English and the Croatian version used in the Croatian study; some food types were difficult to translate into English and were thus left in Croatian). Study participants also gave information about their lifestyle, education, diseases, comorbidities and medications, and signed an informed consent for the anonymous use of collected data (coded questionnaire, biochemical and DNA damage data) following the EU General Data Protection Regulation. Participants were recruited over several months due to limited hospital facilities, but all followed an identical diet regime. For each participant, the study started on the first day of hospitalization, when they signed the informed consent, filled out the questionnaires, anthropometric measurements were taken, and blood was drawn for biochemical and DNA damage analysis. The 3-week 567 kcal VLCD consisted of three daily meals freshly prepared by the hospital nutritionist, with special attention given to hydration conditions (mineral water was readily available during the hospital stay). Participants continued to take their medications and were permitted to take short walks in the hospital park under medical staff surveillance. Patients also performed daily stretching-based exercise of mild intensity under medical staff surveillance. Special attention in meal preparation was given to the carbohydrate–glycemic index and the vitamin and mineral content, with each meal consisting of 50–60% complex carbohydrates, 20–25% proteins, and 25–30% fats (Table 1).

### 2.3. Anthropometric Measurements

For each participant, body weight (kg), body height (m), BMI (kg m^−2^), skeletal muscle mass (SMM, kg), body fat mass (BFM, kg), percentage of body fat (PBF, %), waist–hip ratio (W-HR), visceral fat level (VFL), and basal metabolic rate (BMR, kcal day^−1^) were measured on the first and last day of diet and hospitalization using a body composition analyzer (InBody 270, InBody, Seoul, Korea) according to the manufacturer’s instructions (bioelectrical impedance method). Bray’s method was used to calculate the percentage of excessive body weight loss [22].

### 2.4. Biochemical Parameters

Serum biochemical parameters at the beginning and end of the diet were selected as those expected to be possibly affected during this restricted diet and also connected with DNA (in)stability [23,24,25,26,27].

Blood samples were collected in serum tubes with a silica clot activator (Becton Dickinson, Franklin Lakes, NJ, USA) and used to measure glucose, urea, C-reactive proteins (CRP), total cholesterol (TC), HDL-C, LDL-C, and triglycerides (TG) on a clinical chemistry autoanalyzer Beckman Coulter AU 480 (Beckman Coulter, Inc., Brea, CA, USA). An immunoassay analyzer ARCHITECT i1000sr (Abbott, Chicago, IL, USA) was used to measure insulin, TSH, fT3, and fT4 levels. Insulin resistance as HOMA-IR was calculated as: glucose (mmol L^−1^) × insulin (mIU L^−1^)/22.5. The leukocyte count was determined on an automated hematology analyzer XS-1000i with autosampler (Sysmex, Kobe, Japan).

### 2.5. CBMN Cytome Assay

For the CBMN assay, blood samples collected by venipuncture into heparinized tubes (Beckton Dickinson, Franklin Lakes, NJ, USA) were coded, stored at 4 °C, and transferred within 2 h to IMROH where they were immediately embedded into a cell culture medium. For each sample, duplicates of the cell culture were prepared in a 25 cm^2^ flask through the addition of 0.6 mL heparinized blood under sterile conditions into a prewarmed cell culture medium. The cell culture medium consisted of 6 mL RPMI-1640 medium, 1 mL fetal calf serum (FCS), 0.02 mL phytohemagglutinin-L, and 0.01 mL antibiotic solution (100 IUmL^−1^ penicillin, and 100 mg mL^−1^ streptomycin) and was prepared freshly and kept at 37 °C at least 1 h before cultivation. Cultures were successively incubated at 37 °C, with 5% CO_2_ in the air in a humidified atmosphere in the cell incubator (Thermo Fischer Scientific Inc.-former Heraeus, Langenselbold, Germany).

After 44 h incubation, cytochalasin-B was added to the cultures in sterile conditions at a final concentration of 6 mg mL^−1^ for cytokinesis blocking. After 72 h incubation, cells with the medium were transferred into glass centrifugation tubes, which were run at 450× *g* for 10 min in a centrifuge (Rotofix 32a, Hettich, Tuttlingen, Germany) using a swing bucket rotor. Then, the solution was removed and the cell pellet was mixed gently with a cold mild hypotonic solution (75 mM KCl) and left at room temperature for 10 min. After supernatant removal, the cell pellet was fixed with a fresh mixture of cold methanol/acetic acid (3:1 *v*/*v*) (Kemika, Zagreb, Croatia). The treatment with the fixative was repeated three times, and the cell pellet, dissolved in a minimal volume of fixative, was seeded on clean, cold microscopic slides (Vitrognost, Zagreb, Croatia), dried and stained for 10 min with 5% Giemsa (pH 6.8) prepared freshly in distilled water (Yasenka, Vukovar, Croatia). Microscope analysis was performed at 400× magnification using a light microscope (Olympus, Tokyo, Japan). MNi, NPBs, and NBs were scored in 2000 binucleated lymphocytes with well-preserved cytoplasm per subject. A total of 1000 lymphocytes per donor were scored to evaluate the frequency of cells with 1–4 nuclei (M1, M2, M3, and M4) within the same cytoplasm. The cytokinesis-block proliferation index or nuclear division index (NDI) was calculated according to the following formula, with M1–M4 representing the number of cells with 1–4 nuclei, and N the total number of cells scored (1000) [28].
NDI = (M1 + 2M2 + 3M3 + 4M4) / N

The frequency of apoptotic and necrotic cells in 1000 lymphocytes per subject was also scored, and the scored DNA damage parameter frequency (per 2000 binucleated lymphocytes (BN)) was calculated and also expressed as 1000 BN.

### 2.6. Antioxidative GSH and Oxidative ROS Damage Measurement

As oxidative stress response may affect DNA damage markers, insulin resistance and overall health status during body weight loss, serum ROS and GSH levels were determined and correlated with DNA damage parameters.

#### 2.6.1. GSH Measurement

A monochlorobimane (MBCl) fluorescent probe was used to measure the GSH level. Reaction of MBCl with GSH is highly specific and results in the forming of a fluorescent adduct. A quantity of 100 µL undiluted human serum was pipetted in pentaplicate into a 96-black well plate (Nunc, Thermo Fisher Scientific, Waltham, MA, USA) following the addition of 20 µL 0.24 mM MBCl. Autofluorescence was examined by preparing the serum without the addition of the fluorescent probe. All samples were incubated for 30 min at 37 °C. Fluorescence was determined at the 355 nm excitation and 460 nm emission wavelengths using a microplate reader Victor3™ (Perkin-Elmer, Ynysmaerdy, UK) [29].

#### 2.6.2. ROS Measurement

ROS in human serum were measured using the 2′,7′-dichlorofluorescin diacetate (DCFH-DA) staining assay [30]. Diluted 10% (*v*/*v*) human serum in an ice-cold PBS buffer (pH 7.4) was used by pipetting a volume of 100 µL in pentaplicate into the 96-black well plate (Nunc, Thermo Fisher Scientific, Waltham, MA, USA) following the addition of 20 µL 0.12 mM DCFH-DA in each well. Autofluorescence was examined by preparing the serum without the addition of dye. After 20 min incubation at 37 °C, fluorescence was measured at the 488 nm excitation and 525 nm emission wavelengths using a microplate reader Victor3™ (Perkin-Elmer, Ynysmaerdy, UK).

### 2.7. Statistical Analysis

The Statistica^®^ 13.05.0.17 software package (TIBCO Software Inc., Palo Alto, CA, USA) was used for analysis. Parameters expressed in tables and figures were mean ± standard deviation (SD), standard error (SE), median, and range. Anthropometric, biochemical, DNA damage, and oxidative damage parameters after descriptive statistics were analyzed using Mann–Whitney U-test. The EPIC-Norfolk food questionnaire was analyzed in this study only for daily calorie intake with FETA software to compare the values with the calculated basal metabolism rate [31] before the start of the VLCD. Spearman correlation was used for data correlation, with a significance level set at *p* ≤ 0.05. All data with a decimal point, except for statistical significance, were set to two decimals.

## 3. Results

Demographic information of the study participants including information on gender, age, medications, lifestyle factors, and calorie intake from the EPIC questionnaire are shown in Table 2. The age of the group was (mean ± SD, SE, median, range): 56.12 ± 8.03, 1.57, 58.50, 41–66 years. The group was recruited during 2019 and 2020. Since the group included severely obese patients, it was expected that most, if not all, will have health problems, comorbidities, and diseases, etc. That was the reason for not changing their regular therapy during the diet, and for those for whom it proved a necessity to change it, they were excluded from the study. Among them, only 3 participants received no therapy, 2 participants reported no disease, 16 participants were diabetics, and 23 participants had already developed hypertension. There were two smokers; the first had hypertension, had smoked tobacco for 30 years and reported smoking 300 cigarettes monthly, while the second reported no diseases and had smoked for 4 years—also 300 cigarettes per month. Smokers (*n* = 2) were not permitted to smoke in the hospital or nearby, but could smoke while taking a short walk in the hospital grounds. However, considering they walked usually once per day (if at all), both patients lowered the number of smoked cigarettes per day. The group was characterized by an even sex ratio (13 males and 13 females).

Table 3 provides the group values of the anthropometric and biochemical parameters before and after the 3-week 567 kcal VLCD obtained from all 26 study participants. Calorie values measured and calculated with the bioelectrical impedance method reported in Table 2 served for the comparison of the values calculated from FETA software based on the EPIC questionnaires filled out by participants and reported in Table 3. At the group level, mean values for BMR and calculated calorie intake (CI) at the beginning of the study were similar, though the range of values demonstrated that several individuals did not report true portions and values of what they consumed in the dietary questionnaire, so the questionnaire details were not further analyzed in this study. Group BMI was significantly lower after the diet, while there was a mean decrease in body weight of 8 kg at the group level, or of 10 kg when medians were compared. Waist to hip ratio was also significantly altered after the diet. Other anthropometric factors did not significantly differ after the diet, although all demonstrated lower values (Table 3).

Individual values and differences in anthropometric measurements before and after the diet can be seen in Table 4. The percentage loss of excessive body weight was highly individualized (one person lost almost 35%, one 27%, one 22%, three 20%, two 18%, four between 13 and 14%, two 11%, eight around 10%, one 8%, and one 6%). Before the study, the highest weight was 183.6 kg and the lowest 111 kg. After the diet, the maximum weight was 172.30 and the minimum 105.70 kg. Maximum weight loss was 14.40 kg, and the minimum weight loss was 3.80 kg, with an average loss of 9 kg at the group level. The maximum percentage of body weight loss was 9.60%, and the minimum was 3%, with an average of 6%.

All participants except one outlier had a body fat mass decrease ranging from 2.20 to 8.30. Average PBF loss was 1.22% (range 0–4.40%), and BFM loss was 5% (range 0–8.30%). The BMR difference was a minimum of 28 and maximum of 303 kcal day^−1^, with an average of 90 kcal day^−1^. As for BMI difference, the minimum was 1.50 and maximum was 5, with an average of 3 units (kg m^−2^) for the entire group. Differences for other anthropometric parameters demonstrated once more that weight loss was highly individualized, with differing muscle and fat percentages, again highlighting the strong individual differences.

At the group level (Table 3), lipid profile parameters (TC, HDL-C, LDL-C, and TG) were all significantly lowered after the diet, with TC, LDL-C, and TG levels entering the normal interval reference range. After the diet, the glucose levels were significantly lower and fell into the range of normal values. HOMA-IR was also significantly lower after the diet, whereas insulin levels were on the borderline of significance. Urea levels were also significantly lower after the diet.

The total values of the DNA damage parameters obtained by the CBMN assay are shown in Table 5 for 2000 binucleated cells (BN) per sample, including necrosis on 1000 counted cells and the results of the NDI for each category (M1–M4) on 1000 counted cells. Final statistics were taken or calculated from 1000 cells (Table 5) or 1000 BN cells (DNA damage frequency, Figure 1). The DNA damage frequency results together with significant apoptosis results are shown in Figure 1a (MNi), Figure 1b (NBs), Figure 1c (NPBs), and Figure 1d (apoptosis frequency). All these parameters significantly decreased after the diet at the group level. Almost all parameters of the CBMN *cytome* assay were significantly lower after the diet. Additionally, all parameters of DNA damage significantly declined by nearly half of the initial value. This means that not only damaged cells with the loss of whole/part of the chromosomes were eliminated (decrease in MNi values and the number of BN cells with MNi), but also the number of cells with nuclear buds (amplification of excessive DNA) and nucleoplasmic bridges (cells with dicentric chromosomes) were diminished. The proliferation rate (NDI and M2 cells frequency) increased slightly but not significantly. Necrotic cell frequency slightly decreased, but most importantly, the frequency of cells entering apoptosis (programed cell death) significantly decreased by half after the diet, demonstrating the positive effect of VLCDs on genome stability.

Micronucleus frequency can be affected by smoking status, gender, age, and vitamin concentration, especially vitamin B_12_ and folic acid. As the group contained only two smokers, the smoking factor could not be analyzed. The group consisted of the same number of females and males. The values for vitamin B_12_ and folic acid (measured to check for possible vitamin and mineral deficiency as an exclusion factor—results not shown) did not differ before and after the diet. Importantly, the difference in DNA damage parameters before and after the diet was also confirmed with sex grouping.

Age did not show an effect on DNA damage parameters after the diet, however, differences in DNA damage were observed before the diet (Table 6). When individuals were initially categorized by age into two age group categories of >60 vs. <60, higher MN and NB frequencies were seen in the >60 age group. Interestingly, there were higher levels of NPB in the <60 age group. More detailed categorization of groups by age into categories of 41–50, 51–59, and 60–68 years before the diet showed that the 41–50 category had higher NPB levels compared to the 51–59 and 60–68 groups.

We also compared obtained MN values before and after the diet with the established MN values in the healthy Croatian population (mean ± SD, median, range, and upper highest normal value 12.5 MN per 1000 BN cells). We found that before the diet, 19 individuals had MN values above the median value of seven, while after the diet only four individuals had MN values above the median value.

At the group level, serum GSH levels (mean ± SD, median, range; in RFU units) after the diet decreased slightly (2460.42 ± 672.94, 2500.75, and 378.25–5446.25 vs. 2286.89 ± 698.63, 2225.25, and 1014.25–4686.25), while serum ROS levels (in RFU units) significantly increased after the diet (1003.65±198.36, 993.58, and 701.58–1559.58 vs. 1254.02 * ± 281.69, 1196.58, 784.58–1803.58, and * *p* < 0.05).

Although the study group was small and heterogeneous, a significant positive correlation was found between the pre-diet GSH levels and M1 cell frequency (R = 0.41), and a negative correlation with M2 cell frequency (R = -0.43), confirming that higher GSH levels in obesity negatively affect mitotic cell division. When study participants were divided into two equal-sized groups based on pre-diet GSH, the group with higher GSH levels had slightly lower DNA damage levels and higher apoptotic cell frequencies (Table 7).

Since there were different responses to GSH decreases after the diet and higher GSH values can be caused both genetically and by the type of the food eaten before the VLCD, particular attention was given to those participants who showed reduced serum GSH levels after the diet. There are studies demonstrating that lowering GSH values improves basal metabolism and weight loss. Indeed, 15 study participants had lower GSH levels at the end of the diet, and in general they also had a lower apoptotic frequency (non-significant) and MN frequency (significant) (see Figure 2a,b) when compared to those who did have a reduction in GSH levels after the diet.

Considering CRP, HOMA-IR, or ROS values, the study group was too heterogeneous to draw any clear conclusions. TSH, fT3, and fT4 values were in the reference range both before and after the diet, though TSH and fT3 slightly decreased and fT4 slightly increased after the diet. None of these parameters demonstrated a correlation with BMI, BMR, HOMA-IR, or DNA damage parameters, before or after the diet.

## 4. Discussion

Excessive body weight and obesity are associated with incidences of multiple co-morbidities, such as DMT2, cancer, and cardiovascular disease, that are also major NCDs [5,6,7,31]. Pathophysiological mechanisms underlying obesity are explained with the increase of free fatty acids released from adipose tissue, lipid intermediates, insulin resistance with excess total and intra-abdominal adipose tissue, and inflammation [32,33,34,35,36,37]. Oxidative stress and inflammation, usually present in obesity, induce DNA damage, inhibit DNA damage repair, and cause disturbances in cell metabolism, promoting cancer growth (cancer cell proliferation and migration) and resistance to apoptosis [38].

Obesity thus contributes to hypertension development, cancer pathogenesis (increased lipid concentrations, lipid signaling), and the acceleration of cancer progression (low grade inflammatory response), and is connected with the increased incidence of 13 different cancers [32,33,34,35,36,37,38,39].

Results on more than 120,000 Europeans demonstrated that severe obesity in both genders is associated with the loss of 7 to 10 disease-free years, reflecting excess disease complications and mortality risks in the severely obese that can be lowered by changing the key drivers of an obesogenic environment, such as reduced physical activity, calorie intake, and changes in dietary habits [31]. Even the loss of only excess body fat lowers the risks of most cancers [7].

The results of a meta-analysis within systematic reviews on VLCDs indicated that this type of diet can be more effective than other diets with a less strict calorie restriction in both the short- and long-term, and is more effective than bariatric surgery, demonstrating that a fast weight loss regime was not connected with a faster body weight regain in the period of 12, 24, 36, or even 60 months afterwards [for details see reviews 25,26,40]. VLCDs also demonstrated other superior beneficial outcomes, such as more effective glycemic control, insulin resistance improvement, and improved lipid metabolism [25,26]. VLCDs are generally defined as a very low total energy intake of less than 800 kcal day^−1^ or 3.350 KJ day^−1^ [for details see reviews 25, 26]. With a growing body of studies on the efficacy and acceptability of VLCDs in overweight and obese patients, VLCDs have been considered and confirmed as effective and safe options for weight loss in the treatment of obese patients [26]. However, due to rapid weight loss, VLCDs can cause complications including hypokalemia, cardiac arrhythmia, hyperuricemia, cholelithiasis, and mood and behavior alterations, and therefore should be performed only under strict and even 24 h medical surveillance [40,41].

This is the first study on the assessment of the level of permanent genomic stability parameters in a 3-week hospital-controlled VLCD. Our previous study (Ožvald et al. 2021, under revision) demonstrated the usefulness of the same diet on lowering the level of primary (repairable) DNA damage and primary oxidative DNA damage assessed with alkaline and Fpg alkaline comet assay [42]. This study aimed to evaluate permanent DNA damage during the VLCD using CBMN assay parameters and to assess possible correlations with changes in other anthropometric and biochemical parameters during the diet.

Although the study recruited only 26 individuals, it revealed important results; a short 3-week 567 kcal VLCD in severely obese patients diminished the types of permanent DNA damage values by half. In addition, this diet regime significantly decreased weight as an anthropometric parameter, blood glucose levels, HOMA-IR, TC, HDL-C, LDL-C, TG, and urea as biochemical parameters, and serum GSH levels, not only as an antioxidant but also as a parameter connected with higher metabolism rates, and almost significantly lowered BMI and insulin levels. The study also demonstrated that participants who showed a decrease in GSH concentrations after the diet also had lower DNA damage, while this was not found in participants who did not have a reduction in GSH values.

Our results on anthropometric parameters are comparable, or preferably more beneficial than the results of other diets. Mild restriction diets applied over longer periods (1750–2100 kcal daily, only a 500 kcal daily deficit more than the regular calorie intake in patients) resulted in a 3.5% loss of body weight (2 kg) in obese people, but only after 3 months, in addition to 10% after 6 months or 1 year, and a 17% body weight-loss only after 2 years of the diet [24,43]. For a 2-week 600 kcal day^−1^ VLCD used as a pre-treatment before bariatric surgery, there was a BMI reduction of 1.60 kg m^−2^ (range—0.20 to 3.10), lean body mass loss of 2.80 kg, and fat mass loss of 1.70 kg, with the mean loss of 4.5 kg body weight (range—0.30 to 9.50) and 8.80% of excess body weight (range—0.90 to 17.10) after those 2 weeks [43]. More pronounced weight loss (15–25%) than reported here was demonstrated in VLCD studies after 8 to 16 weeks, and in a 300-kcal day^−1^ VLCD applied for 6–8 weeks in obese people with a BMI > 40, with weight loss of 10% in women (13 kg) and 12% in men (20 kg), along with an improvement of severe clinical conditions, even with the need of reduced dosages of medications, especially diuretics, antihypertensives, and hypoglycemics [39,44,45,46,47]. However, the latter study (Mancini et al.) also reported five patient deaths among the 100 obese subjects who entered the diet program, while some patients stopped the diet due to a worsening of their health conditions, primarily associated with cardiac problems due to the diet [39]. Our VLCD program did not cause any deaths or the need for any participants to withdraw from the study. In addition, changes in medications were made for several participants at the end of the study due to an improvement in their health status, with dose reductions of antihypertensives and antidiabetics. Regarding visceral abdominal fat decreases in obese subjects with DMT2, some VLCD studies have demonstrated a decrease after only 3 days together with body weight reduction [48,49], with further decreases of 25% or 55% after 6 or 16 weeks, respectively, along with a 16% or 45% decrease of abdominal subcutaneous fat, respectively [48,49]. A longer VLCD also demonstrated a significant decrease in the BMI, VFL, and PBF parameters [48,49], which could also be projected for this VLCD program if the diet was to be applied for a longer time (more than 3 weeks). This study included severely obese patients with highly individualized ratios of different fat types and muscle loss, and in this short 3-week period nearly all of the assessed anthropometric parameters were observed to decline at the group level, although the difference was not statistically significant.

Weight loss results in a loss of metabolically active tissue, and therefore decreases BMR [50,51]. BMR gives the number of calories burned to maintain normal function. Different factors such as genetics, age, sex, hormones, and body composition affect BMR. Our diet program decreased BMR in all study participants, though in varying amounts, demonstrating highly individualized differences and no connections with other measured parameters in this study. We found a decrease in lipid parameters (TG, TC, HDL-C, and LDL-C), though with a maintained HDL-C/LDL-C ratio before and after the diet, similarly as reported elsewhere [24]. As for the non-significant decrease in CRPs, Lips et al. demonstrated that a significant decrease in patient obesity can occur only after 3 months on a VLCD [52]. The decrease was nearly 50% of the values before diet. Our CRP baseline and post-diet levels were still higher than the values reported by Lips et al., demonstrating the severity of the inflammatory conditions in our patients (mean ± SD, baseline 7.60 ± 1.20, 3-month VLCD: 4.70 ± 1.30 mg L^−1^ vs. our results: pre-diet: 9.66 ± 8.12, post-VLCD values: 8.62 ± 8.11 mg L^−1^) [52]. Similarly, a non-significant decrease after our diet was observed for the leukocyte count (another inflammatory biomarker).

The impacts of the urea concentrations on multiple inner-organ DNA damage, and with oxidative DNA damage levels, have been demonstrated in other studies [53]. In our previous study with the alkaline comet assay, obese subjects with the highest level of primary DNA damage had also the highest weight and BFM values before the diet and the highest change in the pre- and post-VLCD urea concentrations [42]. This study did not find significant correlations between urea levels before and after the VLCD with the level or decrease of permanent DNA damage.

The slight fT4 increase and TSH and fT3 decreases observed here were also found in other short fasting diets [54]. Increased fT4 but not fT3 levels suggests the presence of undefined factors stopping the conversion of elevated fT4 to fT3 levels in calorie restricted conditions or fasting [54]. We did not find any correlation with the change in other measured parameters in our study, and a similar trend was also observed in our previous study with VLCD and comet assay (Ožvald et al. 2021, under revision) [42].

Similar results in the decrease of glucose, insulin, and HOMA-IR levels in our VLCD were found in another study with obese patients and rapid weight loss with VLCDs [55]. Although glucose levels at the beginning of our study were similar to the values of Lips et al. (8.07 ± 2.78 vs. 8.7 ± 0.5 mmol L^−1^), by taking into account both glucose and insulin levels (16.75 ± 6.98 vs. 11.90 ± 1.50 mIU L^−1^) and calculating HOMA-IR, our group with higher values (6.35 ± 3.99 vs. 4.60 ± 0.60) demonstrated a greater risk for metabolic complications, and although post-diet results were not comparable (our 3-week treatment and their 3-month treatment with lower values than in our study), a significant decrease in all three parameters demonstrated that we managed to lower the risk of complications [52]. In a previous study on patients with DMT2, improved insulin sensitivity was the factor that improved glycemic control [56,57]. Since we recruited patients for this study in 2019 and 2020, after the diet they continued to be included in the program and observed. There was at least one patient who managed to regulate glycemic values after the diet, and the patient’s metformin therapy was cancelled. Even now, 2 years later, this patient continues to control DMT2 without any further medication, only with regular exercise, a prescribed diet, and regular check-ups. This patient has achieved the sustainable remission of DMT2. Most of the other patients’ medications were changed after the study at the individual level, based on the analysis of data created during the study. Although all patients in this study had similar conditions during the VLCD, the final results were highly individualized for each patient, with an individualized approach for each patient determining what they should do and accomplish further after the completion of the diet.

Although obesity can lead to increased estrogen production, causing DNA damage via increased mitotic activity and/or directly via formation of DNA reactive metabolites, studies with similar restriction diets in rats demonstrated that sex hormones do not play a critical role in obesity-related DNA damage, oxidative base damage, or nucleotide excision repair (NER) [23,58]. Therefore, our study was not focused on the sex-related response.

Recent reviews demonstrated an improvement of subjective depressive symptoms following a VLCD program (for details see [59]). Furthermore, VLCDs proved to be a useful approach in the weight treatment of obese patients with different comorbidities such as DMT2, and despite DMT2 being regarded as progressive and incurable, it appeared to be reversible by means of VLCD regimens, as claimed by other authors [26].

Although our previous study with VLCDs using the comet assay and Fpg comet assay demonstrated a decrease in oxidative DNA damage levels, the increased serum ROS values after the VLCD this time were unexpected [42]. A decrease in DNA damage is usually followed by a decrease in ROS production, which was not the case in this study. It is likely that the measured serum ROS cannot be used as a marker of oxidative DNA damage. Due to NDI increase and changes/increases in cell counts after the VLCD (leukocytes, erythrocytes—results not shown in this study), it cannot be excluded that elevated ROS levels are not indicative of oxidative stress, but of higher metabolic activation in the cells connected with cellular repair, etc. It has been demonstrated that if weight loss is induced by an increase in energy expenditure or metabolic rate, there may be an elevation in the mitochondrial production of ROS in the cells, thereby leading to higher ROS levels, which does not reflect the increase in DNA damage [60,61]. The beneficial decrease in GSH levels can be also explained by the elevated ROS levels, and that signaling through ROS, along with other redox events such as thiol oxidation and lipid peroxidation, can also be associated with beneficial health outcomes (for details see review by Picklo [62]).

It seems that GSH is not only a cellular antioxidant that can decrease the level of oxidative stress and be protective against many diseases when present in elevated levels, but a decrease in GSH levels can increase energy expenditure, prevent obesity, and reduce insulin resistance [62]. The extent to which these molecular mechanisms of GSH and ROS contribute to obesity in humans and may be regulated through dietary means needs further research, though it has been demonstrated that elevated GSH reduces insulin sensitivity in adipocytes [62,63]. The positive correlation of a decrease of GSH and decrease of DNA damage after our VLCD program confirms those findings. Our future research will repeat these measurements and use more specific parameters (biomarkers) of oxidative damage and antioxidants to address these observed facts.

Finally, the most important findings of the study were the decreases in DNA damage values. Considering that we analyzed the blood (lymphocytes) that circulated through different organs, the observed DNA damage with CBMN may reflect the overall condition of the body. Although an estimation of DNA damage and DNA repair changes could potentially be a useful biomarker in the risk assessment and prevention of obesity-associated metabolic disorders and cancers at the group level (severely obese) but also at the individual level (susceptible individuals), there are few studies on obese people on a standard diet using parameters of genomic stability, and no studies on VLCDs and permanent genomic (in)stability in obese and severely obese patients.

The recent literature reviews on genomic instability demonstrated elevated levels of DNA damage in overweight, obese, and severely obese people, with different genotoxic techniques (comet assay, micronucleus assay, telomere length, ɣ-H2AX, DNA abasic sites, and 8-oxoguanine measurements, etc.) [14,15,23], but among those studies, few performed an intervention and analyzed the DNA damage levels after normal calorie restriction diets or bariatric surgery. Changes in DNA damage levels (if any) were shown only after 6 months (slight) or after a year of diet or bariatric surgery (for details see reviews [14,15,23]).

The CBMN assay showed its usefulness as a genomic stability biomarker and demonstrated that a short 3-week 567 kcal VLCD can cut the types of permanent damage values by half. Considering that median and mean values of the entire group were above 10 before the diet, with 20 individuals exceeding the mean and median values of the healthy Croatian populations (mean ± SD: 6.90 ± 3.32, median 7 MNi) [64], after the diet only four individuals still had higher values while the mean group level was below the mean healthy values, the CBMN assay demonstrated its value not only at the group level but also in revealing susceptible individuals that should be observed further to establish the behavior of DNA damage after the diet during the program. This highlights our program as a promising personalized approach to resolving the problems associated with obesity.

The value of this study is that other parameters such as NBs and NPBs, and the frequencies of apoptotic and necrotic cells, have not typically been measured in other studies. Only the study of Donmez-Altuntas et al. showed a significantly elevated frequency of apoptotic and necrotic cells as cytotoxicity markers in peripheral blood lymphocytes of obese and total overweight/obese subjects compared with normal-weight subjects, but that study did not use diet intervention [65]. Epidemiological evidence reporting an increased incidence of all cancers for subjects with high CBMN parameters confirm that elevated levels of CBMN parameters play a causal role in cancer development [66,67]. There is increasing evidence that CBMN parameters are linked to the pathogenesis of metabolic and cardiac diseases [15,16] and the severity of coronary artery disease [68,69]. Long-term follow-up studies have confirmed that CBMN is a predictive biomarker of cardiovascular mortality in the healthy population but also in populations with cardiovascular disease and coronary artery disease [70]. Fenech et al. in their review from 2020 demonstrated that the lymphocyte CBMN assay may prove to be a useful tool for the screening of obesity and the metabolic syndrome and its progression to diabetes and CVD in adults [16]. Thus, the CBMN assay can be used at both the group level and individual level, for an individualized or personalized medical approach in treating obesity.

## 5. Conclusions

We demonstrated for the first time that a 3-week 567 kcal hospital-controlled VLCD on obese people with BMI ≥ 35 kg m^−2^ can result in a significant decrease of half of the values of DNA damage parameters measured by the CBMN assay, and a significant decrease of glucose, urea, TC, HDL-C, LDL-C, body weight, GSH, and insulin levels, as well as the percentage of excessive body weight and an improvement (almost significantly) in BMI by 3 or even 5 units. Although the group was highly heterogeneous, consisting of patients with class II and III obesity, which can explain the lack of a significant difference in post-diet BMI (although there was a decrease), the results on DNA damage and selected biochemical parameters are relevant for the development of healthy weight loss strategies. A recent review characterized DNA damage in obesity as an initiator, promoter, and predictor of cancer and advocated for early, pre-malignant assessment of genome integrity and stability to develop surveillance strategies and interventions [71]. Further studies will be conducted on a higher number of volunteers, and will include other biomarkers, especially biomarkers of oxidative stress and antioxidants, with continued surveillance of subjects who have already entered the program. Special attention will be also given to informing patients about the inflammatory potential of their habitual diet and the Dietary Inflammatory Index of specific foods, and to analyze the possible connections of these foods with the results [72]. These results will facilitate the further development of the program at the Duga Resa Special Hospital for Extended Treatment so that in the future it will also include 50 g of high-quality proteins and 55 g of carbohydrates per day, and adequate vitamin and mineral supplementation regimens during the VLCD.

## Figures and Tables

**Figure 1 nutrients-13-04468-f001:**
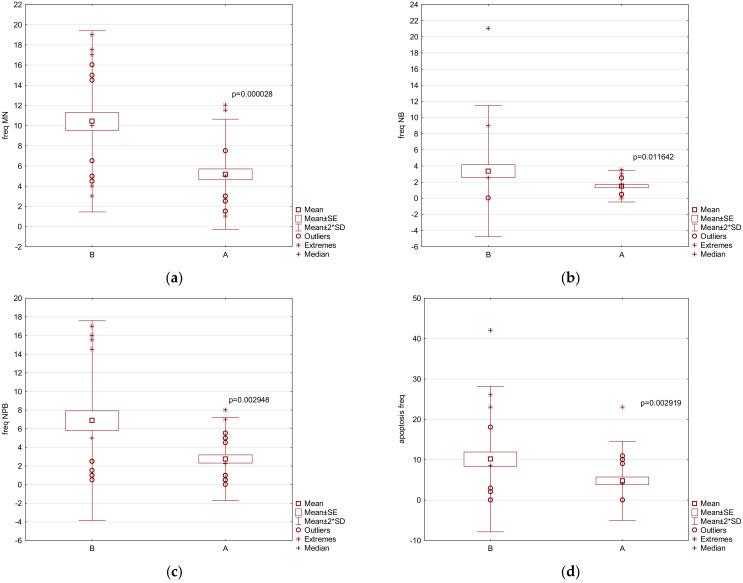
Results of CBMN parameter frequency before (B) and after (A) the 3-week 567 kcal VLCD in the study group of obese patients (*n* = 26, BMI ≥ 35 kg m^−2^). Frequency was calculated from 2000 to 1000 binucleated cells (BN): (**a**) MNi-micronuclei; (**b**) NBs-nuclear buds; (**c**) NPBs-nucleoplasmic bridges; and (**d**) frequency of apoptotic cells per 1000 cells.

**Figure 2 nutrients-13-04468-f002:**
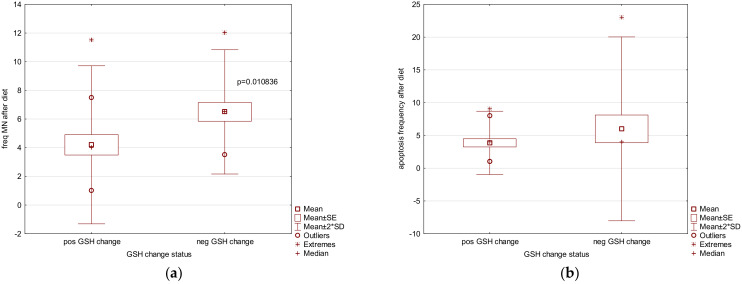
Positive GSH change (decrease, contrary to a negative GSH change as an increase in GSH values after the diet) in reduced glutathione (GSH) values after the 3-week 567 kcal VLCD in the study group of obese patients (*n* = 26, BMI ≥ 35kg m^−2^) demonstrated an influence on: (**a**) significantly lower frequency of MNi (per 1000 BN cells); (**b**) lower frequency of apoptotic cells (per 1000 cells).

**Table 1 nutrients-13-04468-t001:** The composition of the 567 kcal hospital-controlled VLCD (main dietary components).

35.91 g proteins	-
15.05 g fats	5.05 g saturated, 4.55 g monounsaturated fatty acids, 4.67 g polyunsaturated fatty acids, 140.42 mg cholesterol
74.92 g carbohydrates	18.46 g monosaccharides and disaccharides, 56.46 g polysaccharides, 14.09 g fibers
Minerals	1144.68 mg Na^+^, 1447.47 mg K^+^, 376.02 mg Ca^2+^, 99.84 mg Mg^2+,^ 7.82 mg Fe, and 636.53 mg P
Vitamins	0.51 mg vitamin B_1_, 0.67 mg vitamin B_2_, 0.57 mg vitamin B_6_, 75.97 mg vitamin C

**Table 2 nutrients-13-04468-t002:** Demographic information (gender, age in years, smoker, existing disease, current therapies, and former tumor incidence) of individuals in the study group (*n* = 26) with calculated calorie intake (CI, kcal day^−1^) from the EPIC questionnaire with FETA software.

Code	G	Age	S	Disease	Therapy	Tumor Incidence	CI
1	f	62	No	none	none	No	1098
2	m	56	No	Hypertension, 10 y, hyperlipoproteinemia	Torasemide 10 mg 1 tbl, nebivolol 5 mg 1 tbl, lercanidipine 20 mg 1 tbl, moxonidine 0.4 mg 1 tbl, perindopril 8 mg 1 tbl, fenofibrate/simvastatin 145/20 mg 1 tbl	No	2287
3	f	64	No	DMT2, 20 y and hypertension 20 y, hyperlipoproteinemia	Metformin 1000 mg 2 × 1 tbl, glargine 32 units, ramipril 5 mg 1 tbl, lacidipine 6 mg 1 tbl, moxonidine 0.2 mg 1 tbl, fenofibrate/simvastatin 145/40 mg 1 tbl	Thyroid tumor 10 y ago	2387
4	m	48	No	Asthma, 20 y, hypertension 20 y, hyperlipoproteinemia	Torasemide 5 mg 1 tbl, perindopril/indapamide 10/2.5 mg 1 tbl	No	2188
5	m	66	No	DMT2, 17 y, hypertension, hyperlipoproteinemia	Perindopril/indapamide/amlodipine 10/2.5/5 mg 1 tbl, empagliflozin/metformin 12.5/1000 mg 1 tbl, dulaglutide 1.5 mg s.c. once weekly, fenofibrate/simvastatin 145/40 mg 1 tbl, torasemide 10 mg 1 tbl every second day	No	497
6	f	65	No	Asthma, hypertension, hyperlipoproteinemia	Furosemide 500 mg ¼ tbl, rosuvastatin 10 mg 1 tbl, losartan 100 mg 1 tbl	No	1888
7	m	42	No	Hypertension, 10 y	Torasemide 10 mg 1 tbl, perindopril 8 mg 1 tbl, moxonidine 0.4 mg 1 tbl, nebivolol 5 mg 1 tbl	No	2283
8	m	66	No	Hypothyroidism, 12 y ago, hypertension, hyperlipoproteinemia	Perindopril 10/10 mg 1 tbl, indapamide 1.5 mg 1 tbl, moxonidine 0.6 mg 1 tbl, nebivolol 5 mg ¼ tbl	Thyroid carcinoma 12 y ago	1311
9	m	51	No	None	None	No	1622
10	f	61	No	DMT2,14 y, and hypertension, 14 y, and anemia, 2 y	Furosemide 40 mg 1 tbl, gliclazide 60 mg 1 tbl, pioglitazone 30 mg 1 tbl, exenatide once weekly, metformin 1000 mg 2 × 1 tbl, perindopril 10 mg 1 tbl, atorvastatin 20 mg 1 tbl, bisoprolol 5 mg 1 tbl	Colon carcinoma 14 y and nose tumor 4 y ago	1754
11	f	52	Yes	Hypertension, DMT2, 10 y, hyperlipoproteinemia	Empagliflozin/metformin 12.5/1000 mg 1 tbl, metformin 1000 mg 1 tbl, perindopril/amlodipine 5/5 mg 1 tbl	No	829
12	m	55	No	Chronic sinusitis, psoriasis vulgaris, arthritis psoriatica, DMT2, 4 y, hypertension, hyperlipoproteinemia	Ramipril 5 mg 1 tbl, lacidipine 4 mg 1 tbl, nebivolol/hydrochlorothiazide 5/12.5 mg ½ tbl, fenofibrate/simvastatin 145/40 mg 1 tbl, metformin 1000 mg 2 × 1 tbl	No	1044
13	f	56	No	Asthma, DMT2, 6 y, hypertension, hyperlipoproteinemia	Furosemide 40 mg 1 tbl, linagliptin 5 mg 1 tbl	Uterus myoma	2030
14	f	67	No	DMT2, 10 y, hypertension, hyperlipoproteinemia	Perindopril/indapamide 5/1.25 mg 1 tbl, metformin 850 mg 2 × 1 tbl	No	2828
15	m	57	No	DMT2, 5 y, hypertension, hyperlipoproteinemia	Perindopril/indapamide/amlodipine 10/2.5/5 mg 1 tbl, metformin 1000 mg 2 × 1 tbl	No	968
16	f	46	No	DMT2, 13 y, hypertension, hyperlipoproteinemia	Ramipril/amlodipine 10/10 mg 1 tbl, torasemide 5 mg 1 tbl, insulin aspart/protamine 30 2× daily, empagliflozin/metformin 12.5/1000 mg, metformin 1000 mg 1 tbl after dinner, rosuvastatin 10 mg 1 tbl.	No	14,656
17	m	50	No	DMT2, 20 y, hypertension, hyperlipoproteinemia	Furosemide 500 mg ¼ tbl, bisoprolol 2.5 mg 1 tbl, atorvastatin 20 mg 1 tbl, perindopril/indapamide 4/1.25 mg 1 tbl, insulin aspart/protamine 30 2× daily, insulin aspart with lunch, dulaglutide 1.5 mg once weekly, dapagliflozin 10 mg 2 × 1 tbl.	No	1228
18	m	68	No	DMT2, 15 y, hypertension, hyperlipoproteinemia	Gliclazide 60 mg 1 tbl, dulaglutide 1.5 mg once weekly, dapagliflozin/metformin 5/1000 mg 2 × 1, pioglitazone 30 mg, metformin 1000 mg 1 tbl after dinner, rosuvastatin 20 mg, perindopril/indapamide 8/2.5 mg	No	1001
19	m	59	No	DMT2, 15 y, hypertension 15 y, osteoarthritis	Furosemide 40 mg 2–3× in week, atorvastatin 20 mg, gliclazide 60 mg ½ tbl, dulaglutide 1.5 mg once weekly, metformin 1000 mg 2 × 1 tbl, perindopril/amlodipine 10/5 mg 1 tbl, bisoprolol 5 mg 1 tbl	No	1588
20	f	61	No	Hyperlipoproteinemia (new diagnosis)	None	Colon 14 y ago and nose carcinoma 4 y ago	1581
21	m	41	Yes	DMT2, 20 y and hypertension 20 y, hyperlipoproteinemia	Dulaglutide 1.5 mg once weekly, glargine 30 j s.c., metformin 2000 mg, furosemide 40 mg 1 tbl, perindopril/indapamide/amlodipine 8/2.5/10 mg 1 tbl, moxonidine 0.2 mg 1 tbl, fenofibrate/simvastatin 145/40 mg 1 tbl	Fibroadenoma	1538
22	f	64	No	DMT2, 20 y and hypertension 20 y, hyperlipoproteinemia	Dulaglutide 1.5 mg once weekly, glargine 32 j in 21 h, metformin 1000 mg 2 × 1 tbl, levothyroxine 175 µg 1 tbl, furosemide 40 mg 1,0,0, KCl 500 mg 1,0,0, perindopril/indapamide/amlodipine 8/2.5/10 mg 1 tbl, moxonidine 0.2 mg 0,1,0, fenofibrate/simvastatin 40/145 mg 0,0,1, pantoprazole 40 mg 1,0,0, fluoxetine 20 mg 1 tbl, oxazepam 10 mg 0,0,1, cholecalciferol D3 8 drops, Mg 375 mg, beclometasone/formoterol 100/67 µg pp, lactulose sol 15 mL pp, trimetazidine MR tbl 35 mg 2 × 1.	Thyroid tumor 10 y ago	1402
23	f	64	No	DMT2, 20 y and hypertension 20 y, hyperlipoproteinemia	semaglutide 1 mg s.c. once weekly, glargine 35 j in 21 h, metformin 1000 mg 2 × 1 tbl, levothyroxine 175 µg 1 tbl, montelukastum 10 mg 1 tbl, desloratadine 1 tbl, furosemide 40 mg 1,0,0, KCl 500 mg 1,0,0, perindopril/indapamide/amlodipine 8/2.5/10 mg 1 tbl, moxonidine 0.2 mg 0,1,0, fenofibrate/simvastatin 40/145 mg 0,0,1, pantoprazole 40 mg 1,0,0, fluoxetine 20 mg 0,0,1, oxazepam 10 mg 0,0,1, cholecalciferol D3 8 drops, Mg 375 mg, beclometasone/formoterol 2 inh100/67 µg pp, lactulose sirup 15 mL pp; trimetazidine MR tbl 35 mg 2 × 1.	Thyroid tumor 10 y ago	1488
24	m	42	No	Hypertension, hyperlipoproteinemia, hyperuricemia	nebivolol 5 mg 1 tbl, torasemide 10 mg 1 tbl weekly, allopurinol 100 mg 1 tbl.	No	1978
25	f	58	No	DMT2, Hypertension, Coronary artery disease, Atrial fibrillation, post coronary angioplasty and stents, hyperlipoproteinemia, hyperuricemia	dabigatran 110 mg 2 × 1 tbl, bisoprolol COR 2.5 mg 2 × 1 tbl, valsartan/hydrochlorothiazide 160/12.5 mg 1 tbl, clopidogrel 75 mg 1 tbl, furosemide 40 mg 1 tbl, KCl 500 mg 1 tbl, amiodarone 200 mg 1 tbl rosuvastatin 10 mg 1 tbl, metformin XR 750 mg 1 tbl.	No	1935
26	f	60	No	DMT2, Hypertension, hyperlipoproteinemia, Hypothyreosis	semaglutide 1 mg s.c. once weekly, metformin 1000 mg 1 tbl, levothyroxine 150 µg 1/2 tbl, febuxostat 120 mg 1 tbl, perindopril/indapamide/amlodipine 5/1.25/10 mg 1 tbl, bisoprolol 2.5 mg 1 tbl, atorvastatin 20 mg 1 tbl	No	1651

G—gender; f—female; m—male; S—smokers; DMT2—Diabetes mellitus type 2; y—years; tbl—tablets; and CI—calorie daily intake (kcal day^−1^) from the EPIC food questionnaire calculated with FETA software.

**Table 3 nutrients-13-04468-t003:** Anthropometric and biochemical parameter values before and after the 3-week 567 kcal VLCD in the study group of obese patients (*n* = 26, BMI ≥ 35 kg m^−2^).

	Reference Intervals (Range)		Baseline/Week 0 Mean ± SD, SE, Median, Range	Post-Diet/Week 3 Mean ± SD, SE, Median, Range	*p*-Value
**Anthropometric measures**	**Weight**/kg		**134.86 ± 16.61, 3.30, 131.15, 111–183.60**	**126.48 ± 15.81, 3.10, 121.75,** **105.70–** **172.30**	***p* = 0.029255 ***
	**W-HR**/ratio		**1.04 ± 0.12, 0.02, 1.06,** **0.70–** **1.28**	**1.10 ± 0.09, 0.02, 1.09,** **0.94–** **1.29**	***p* = 0.046439 ***
	**BMI**/kg m^−2^		47.43 ± 6.00, 1.18, 49.45, 35.70–57.50	44.49 ± 5.80, 1.14, 46.05, 33.60–54.90	*p* = 0.057755
	**SMM**/kg		40.39 ± 7.19, 1.41, 38.05, 30.5–54.20	37.90 ± 7.00, 1.37, 36.35, 28.60–52.40	*p* = 0.146747
	**BFM**/kg		63.74 ± 12.46, 2.44, 64.90, 40.40–90.30	59.50 ± 12.55, 2.46, 61.35, 35–82	*p* = 0.211910
	**PBF**, %		46.83 ± 7.53, 1.48, 46.35, 33.50–64.50	46.07 ± 8.16, 1.60, 47.50, 30.70–62.70	*p* = 0.737078
	**BMR**/kcal day^−1^		1906.04 ± 254.20, 49.85, 1817.50, 1551–2385	1816.88 ± 246.67, 48.38, 1764.50, 1484–2320	*p* = 0.157311
	**VFL**	0–35 range	19.58 ± 1.10, 0.22, 20, 16–20	19.58 ± 1.27, 0.25, 20, 15–20	*p* = 0.9
	**Calorie intake**/kcal day^−1^		2117.65 ± 2614.28, 512.70, 1605.00, 497–14,656-		
**Biochemical measures**	**Glucose**	4.40–6.40 mmol L^−1^	**8.07 ± 2.78, 0.54, 7.50,** **5.30–** **19.10**	**6.54 ± 1.79, 0.35, 5.50,** **4.80–** **11.50**	***p* = 0.004524 ***
	**Urea**	2.80–8.30 mmol L^−1^	**7.03 ± 3.32, 0.65, 5.95,** **3.90–** **18.70**	**5.51 ± 2.72, 0.53, 4.70,** **2.40–** **12.40**	***p* = 0.013070 ***
	**TC**	<5.00 mmol L^−1^	**5.42 ± 1.45, 0.28, 5.30,** **3.00–** **10.10**	**4.08 ± 1.32, 0.26, 3.95,** **2.30–** **6.60**	***p* = 0.001851 ***
	**HDL-C**	>1.20 mmol L^−1^	**1.23 ± 0.35, 0.07, 1.15,** **0.80–** **2.10**	**0.99 ± 0.27, 0.05, 0.90,** **0.60–** **1.60**	***p* = 0.010357 ***
	**LDL-C**	<3.00 mmol L^−1^	**3.30 ± 1.29, 0.25, 3.10,** **1.00–** **7.70**	**2.29 ± 1.13, 0.22, 2.25,** **0.80–** **4.60**	***p* = 0.004262 ***
	**TG**	<1.70 mmol L^−1^	**2.18 ± 0.79, 0.15, 2.10,** **0.80–** **3.70**	**1.72 ± 0.59, 0.12, 1.65,** **0.70–** **3.00**	***p* = 0.029221 ***
	**Insulin**/mIU L^−1^		16.75 ± 6.98, 1.37, 16.35, 4.30–31.20	13.35 ± 7.77, 1.52, 11.59, 2.30–31.50	*p* = 0.058176
	**HOMA-IR**/molar units		**6.35 ± 3.99, 0.78, 5.05, 1.40–** **17.70**	**4.08 ± 3.33, 0.65, 3.05, 0.50–** **16.10**	***p* = 0.007332 ***
	TSH	0.35–4.94 mIU L^−1^	1.91 ± 1.13, 0.22, 1.68, 0.08–5.72	1.38 ± 0.92, 0.18, 1.15, 0.03–3.24	*p* = 0.131075
	fT4	9.00–19.05 pmol L^−1^	13.70 ± 2.00, 0.39, 13.85, 9.48–18.57	15.75 ± 3.82, 0.75, 14.06, 11.61–25.93	*p* = 0.080455
	fT3	2.63–5.70 pmol L^−1^	3.99 ± 0.81, 0.16, 4.00, 2.64–5.59	3.87 ± 0.74, 0.14, 3.88, 2.53–5.11	*p* = 0.682996
	leuk	3.40–9.70 × 10^9^ L^−1^	8.07 ± 2.23, 0.44, 7.31, 5.06–14.13	7.02 ± 1.56, 0.31, 6.92, 4.73–11.11	*p* = 0.113943
	CRP	<5.00 mg L^−1^	9.66 ± 8.12, 1.59, 6.63, 1.68–31.16	8.62 ± 8.11, 1.59, 7.00, 0.68–29.93	*p* = 0.495734

W-HR—waist–hip ratio; SMM—skeletal muscle mass; BFM—body fat mass; PBF—percent body fat; BMR—basal metabolic rate; VFL—visceral fat level; TC—total cholesterol; TG—triglycerides; fT3—free triiodothyronine; fT4—free thyroxine; TSH—thyroid-stimulating hormone or thyrotropin; leuk—leukocytes; CRP—C-reactive protein; HOMA-IR—homeostatic model assessment of insulin resistance; and *—statistically significant difference, *p* < 0.05, Mann–Whitney U-test. Statistically significant differences before and after the diet on the group level are marked in bold.

**Table 4 nutrients-13-04468-t004:** Individual anthropometric parameter values before and after the 3-week 567 kcal VLCD in the study group of obese patients (*n* = 26, BMI ≥ 35 kg m^−2^).

n	Weight/kg	SMM (diff)	BFM (diff)	PBF (diff)	BMI (diff)	BMR (diff)	W-HR (diff)	VFL (diff)
1	128.60→121.60 (7, 5.44%, >10%)	34.90→33.80 (1.10)	66.90→61.30 (5.60)	52.10→50.40 (1.70)	50.90→48.10 (2.80)	1702→1673 (29)	0.92→0.98 (−0.06)	20→20 (0)
2	154.70→140.30 (14.40, 9.30%, ~20%)	52.20→48.30 (3.90)	63.4→56.20 (7.20)	41.00→40.1 (0.90)	54.20→49.10 (5.10)	2341→2150 (191)	0.84→0.99 (−0.15)	19→20 (−1)
3	143.50→131.60 (11.90, 8.30%, ~20%)	38.30→33.50 (4.80)	76.40→71.60 (4.80)	53.20→54.40 (−1.20)	56.10→51.40 (4.70)	1820→1666 (154)	0.70→1.03 (−0.33)	16→20 (−4)
4	133.60→125.80 (7.80, 5.80%, ~13%)	44.40→41.30 (3.10)	55.90→53.00 (2.90)	41.80→42.20 (−0.40)	41.70→39.30 (2.40)	2049→1942 (107)	1.14→1.19 (−0.05)	20→20 (0)
5	145.20→131.30 (13.90, 9.60%, ~20%)	46.40→38.00 (8.40)	64.20→64.30 (−0.10)	44.20→49.00 (−4.80)	50.20→45.40 (4.80)	2120→1817 (303)	1.01→1.25 (−0.24)	20→20 (0)
6	126.50→121.70 (4.80, 3.80%, ~10%)	31.80→30.80 (1.00)	69.80→66.60 (3.20)	55.20→54.70 (0.50)	50.00→48.10 (1.90)	1594→1560 (34)	1.07→1.08 (−0.01)	20→20 (0)
7	183.60→172.30 (11.30, 6.20%, >10%)	54.20→52.40 (1.80)	90.30→82.00 (8.30)	49.20→47.60 (1.60)	54.80→51.40 (3.40)	2385→2320 (65)	1.15→1.23 (−0.08)	20→20 (0)
8	118.70→111.90 (6.80, 5.70%, ~13%)	44.90→43.20 (1.70)	40.50→36.60 (3.90)	34.10→32.70 (1.40)	37.50→35.30 (2.20)	2060→1997 (63)	1.15→1.15 (0.00)	19→18 (1)
9	150.70→141.50 (9.20, 6.10%, >10%)	49.30→47.30 (2.00)	65.60→59.70 (5.90)	43.50→42.20 (1.30)	49.80→46.70 (3.10)	2209→2136 (73)	1.02→1.07 (−0.05)	20→20 (0)
10	131.60→120.70 (10.90, 8.30%, ~22%)	35.80→31.80 (4.00)	66.80→63.20 (3.60)	50.80→52.30 (−1.50)	46.60→42.80 (3.80)	1769→1613 (156)	0.96→1.09 (−0.13)	20→20 (0)
11	145.30→138.80 (6.50, 4.50%, >10%)	37.80→36.30 (1.50)	78.40→74.00 (4.40)	54.00→53.30 (0.70)	57.50→54.90 (2.60)	1815→1770 (45)	1.01→0.94 (0.07)	20→20 (0)
12	128.00→117.30 (10.70, 8.40%, ~27%)	48.00→45.90 (2.10)	43.80→36.80 (7.00)	34.30→31.40 (2.90)	41.30→37.90 (3.40)	2188→2109 (79)	0.91→0.97 (−0.06)	16→15 (1)
13	137.30→131.60 (5.70, 4.20%, ~10%)	34.00→32.30 (1.70)	76.70→73.90 (2.80)	55.90→56.20 (−0.30)	51.70→49.50 (2.20)	1679→1616 (63)	1.04→1.11 (−0.07)	20→20 (0)
14	111.00→105.70 (5.30, 4.80%, 11%)	30.50→28.60 (1.90)	56.30→54.10 (2.20)	50.80→51.20 (−0.40)	45.00→42.90 (2.10)	1551→1484 (67)	0.96→1.00 (−0.04)	20→20 (0)
15	130.70→120.50 (10.20, 7.80%, <22%)	40.30→37.70 (2.60)	59.70→54.10 (5.60)	45.70→44.90 (0.80)	44.70→41.20 (3.50)	1903→1804 (99)	1.17→1.20 (−0.03)	20→20 (0)
16	139.10→132.80 (6.30, 4.50%, 10%)	37.60→36.40 (1.20)	72.90→68.50 (4.40)	52.40→51.60 (0.80)	50.50→48.20 (2.30)	1799→1759 (40)	1.02→1.08 (−0.05)	20→20 (0)
17	120.80→114.10 (6.70, 5.60%, 18%)	46.00→44.80 (1.20)	40.40→35.00 (5.40)	33.50→30.70 (2.80)	35.70→33.70 (2.00)	2106→2078 (28)	1.10→1.09 (0.01)	19→16 (3)
18	121.70→114.70 (7.00, 5.80%, ~11%)	33.00→32.00 (1.00)	62.60→57.50 (5.10)	51.40→50.10 (1.30)	43.60→41.10 (2.50)	1647→1606 (41)	1.16→1.21 (−0.05)	20→20 (0)
19	119.40→110.20 (9.20, 7.70%, <35%)	41.50→39.00 (2.50)	45.90→40.80 (5.10)	38.40→37.00 (1.40)	36.40→33.60 (2.80)	1958→1869 (89)	1.13→1.18 (−0.05)	20→20 (0)
20	116.50→109.40 (7.10, 6.10%, >13%)	33.90→31.50 (2.40)	54.70→51.90 (2.80)	47.00→47.40 (−0.40)	41.30→38.80 (2.50)	1704→1613 (91)	0.94→1.08 (−0.14)	20→20 (0)
21	140.40→129.20 (11.20, 8.00%, <22%)	48.30→45.80 (2.50)	56.60→49.40 (7.20)	40.30→38.30 (2.00)	43.80→40.30 (3.50)	2180→2093 (87)	1.07→1.05 (0.02)	20→20 (0)
22	133.60→126.40 (7.20, 5.40%, >10%)	35.60→33.50 (2.10)	70.00→66.50 (3.50)	52.40→52.60 (−0.20)	52.20→49.40 (2.80)	1744→1664(80)	0.89→1.01 (−0.12)	20→20 (0)
23	125.60→121.80 (3.80, 3.00%, 6.16%)	33.30→31.90 (1.40)	66.60→64.90 (1.70)	43.60→42.10 (1.50)	49.10→47.60 (1.50)	1645→1598(47)	1.08→1.05 (0.03)	20→20 (0)
24	170.50→163.40 (7.10, 4.20%, 8.18%)	51.00→49.60 (1.40)	82.60→78.00 (4.60)	64.50→62.70 (1.80)	50.90→48.80 (2.10)	2269→2216(53)	1.27→1.29 (−0.02)	20→20 (0)
25	127.20→117.80 (9.40, 7.40%, 14.34%)	33.50→30.50 (3.00)	68.00→63.70 (4.30)	43.90→39.90 (4.00)	51.60→47.80 (3.80)	1648→1538(110)	1.14→1.19 (−0.05)	20→20 (0)
26	122.50→112.60 (9.90, 8.10%, 17.66%)	33.60→30.40 (3.20)	62.20→58.10 (4.10)	44.50→40.10 (4.40)	46.10→42.40 (3.7)	1672→1548(124)	1.07→1.16 (−0.09)	20→20 (0)

Mean difference before and after is in brackets. Weight values were expressed as: diff, %weight loss, %excess body weight loss; diff—subtraction values (before–after); SMM—skeletal muscle mass; BFM—body fat mass; PBF—percent body fat; BMI—body mass index; BMR—basal metabolic rate; W-HR—waist–hip ratio; and VFL—visceral fat level.

**Table 5 nutrients-13-04468-t005:** CBMN assay parameter values before and after the 3-week 567 kcal VLCD in the study group of obese patients (*n* = 26, BMI ≥ 35 kg m^−2^).

Parameter	Before	After	*p* Value
	Mean ± SD (SE), Median, Range	Mean ± SD (SE), Median, Range	
**MN1**	18.69 ± 9.04 (1.77), 18, 6–35	**9.50 ± 5.05 (0.99), 9, 2–22**	
**No BN with MN**	19.62 ± 9.02 (1.77), 19.50, 6–35	**9.88 ± 5.18 (1.02), 9.50, 2–23**	
**MNi total**	20.85 ± 8.97 (1.76), 20, 6–38	**10.35 ± 5.47(1.07), 10, 2–24**	
**BN with 1NB**	5.81 ± 5.69 (1.11), 4.50, 0–28	**3.00 ± 1.96(0.38), 3, 0–7**	
**BN with >1NB**	0.46 ± 1.42 (0.28), 0, 0–7	**0**	
**No BN with NB**	6.27 ± 6.87 (1.35), 5, 0–35	**3.00 ± 1.96(0.38), 3, 0–4**	
**NBs total**	6.73 ± 8.13 (1.59), 5, 0–42	**3.00 ± 1.96(0.38), 3, 0–7**	
**NPBs total**	13.73 ± 10.72 (2.10), 10, 1–34	**5.50 ± 4.46 (0.87), 4.5, 0–16**	
M1	716.92 ± 223 (43.73), 826, 197–912	681.85 ± 191.33 (37.52), 743, 260–922	0.301108
M2	223.58 ± 139.86 (27.43), 155, 84–504	262.92 ± 138.73 (27.21), 244, 72–509	0.260328
M3	25.08 ± 32.54 (6.38), 9, 1–97	25.23 ± 26.12 (5.12), 14, 2–103	0.713974
M4	33.62 ± 59.71 (11.71), 7, 0–214	29.23 ± 37.26 (7.31), 8, 0–136	0.790038
NDI	1.37 ± 0.37 (0.07), 1.2, 1.09–2.32	1.40 ± 0.28 (0.06), 1.30, 1.09–2.12	0.420625
Freq necrosis	5.65 ± 9.94 (1.95), 0, 0–33	4 ± 6.58 (1.29), 0, 0–22	0.950682

MN—micronucleus; MNi—micronuclei; BN—binucleated cell; MN1—BN with one MN; NBs—nuclear buds; 1NB—one NB; NPBs—nucleoplasmic bridges; NPB1—NB with one NPB; M1—mononuclear cells; M2—binuclear (BN) cells; M3—three-nuclear cells; M4—tetranuclear cells; and NDI—nuclear division index. Statistically significant results are in bold, here most of them are represented on 2000 BN, and in the Figure 1a–d on 1000 BN with final numbers and statistical significance, not to repeat the results of statistics.

**Table 6 nutrients-13-04468-t006:** Differences in MN, NB, NPB, and apoptotic frequency in different groups before the diet by age category.

	MN	NB	NPB	Apoptotic Frequency
Mean ± SD, SE, Median, Range				
Two age categories				
>60, *n* = 12	11.29 ± 4.59, 1.33, 10.75, 3–17.50	4.46 ± 5.71, 1.65, 2.50, 0.50–21	6.46 ± 5.36, 1.55, 4.25, 0.50–16	8.75 ± 8.09, 2.34, 6.50, 0–26
<60, *n* = 14	9.68 ± 4.42, 1.18, 10, 4–19	2.43 ± 1.47, 0.39, 0–5	7.21 ± 5.54, 1.48, 6, 1–17	11.36 ± 9.86, 2.63, 10, 2–42
Three age categories				
41–50 y, *n* = 6	10 ± 4.32, 1.77, 11.50, 4.50–14.50	2.92 ± 1.80, 0.74, 2.75, 1–5	9.25 ± 6.69, 2.73, 9.50, 1–17	9.50 ± 5.89, 2.40, 10, 2–18
51–59 y, *n* = 8	9.44 ± 4.78, 1.69, 8.75, 4–19	2.06 ± 1.15, 0.41, 2, 0–3.50	5.69 ± 4.33, 1.53, 4.25, 1.50–14.50	12.75 ± 12.27, 4.43, 10, 3–42
60–68 y, *n* = 12	11.29 ± 4.59, 1.33, 10.75, 3–17.50	4.46 ± 5.71, 1.65, 2.50, 0.50–21	6.46 ± 5.36, 1.55, 4.25, 0.5–16	8.75 ± 8.09, 2.34, 6.50, 0–26

MN—micronuclei; NB—nuclear bud; NPB—nucleoplasmic bridge; SD—standard deviation; SE—standard error; and y—years.

**Table 7 nutrients-13-04468-t007:** Differences in MN, NB, NPB, and apoptotic frequency in different groups before the diet by GSH level.

	MN	NB	NPB	Apoptotic Frequency
Mean ± SD, SE, Median, Range				
Higher (H)/Lower (L) GSH levels				
H, *n* = 13	8.69 ± 4.07, 1.13, 10, 3–15	2 ± 1.59, 0.44, 1.50, 0–5	5.27 ± 4, 1.11, 4, 1–14.50	12.15 ± 11.06, 8, 3.07, 3–42
L, *n* = 13	12.15 ± 4.35, 1.21, 12, 6.5–19	4.73 ± 5.27,1.46, 3, 1.5–21	8.46 ± 6.19, 1.72, 8.50, 0.50–17	8.15 ± 6.16, 1.71, 12, 0–23

MN—micronuclei; NB—nuclear bud; NPB—nucleoplasmic bridge; GSH—reduced glutathione; SD—standard deviation; and SE—standard error.

## Data Availability

Individual data are already presented in the manuscript, other data are not applicable due to GDPR, hospital rules, and written patient consents.

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
