# Peer review of "Effects of a 3-Week Hospital-Controlled Very-Low-Calorie Diet in Severely Obese Patients"

_nutrients, 2021, doi:10.3390/nu13124468_

Round 1
Reviewer 1 Report
Obesity which is frequently accompanied by elevated blood pressure, elevated plasma glucose levels, and high serum triglycerides, is associated with the risk of developing cardiovascular disease, type 2 diabetes, and cancer. Increased oxidative stress observed in obesity disturbs cell metabolism and causes DNA damage. Diet and exercise are essential in the treatment of obesity. The manuscript presented by the authors describes how the very low caloric diet within 3 weeks, conducted under hospital control, affects the lipid profile, insulin resistance, oxidative stress, and DNA stability. VLCD seems to be an effective and safe weight loss option in the treatment of obese patients and a very interesting alternative to bariatric surgery fraught with serious complications.
There are several points that authors should correct further improve this manuscript:
- Add information on the total number of patients included in the study, in section 2.2 Subject recruitment and diet.
- Line 83: NBs and in line 172: NBUDs do both abbreviations refer to Nuclear Buds? Should be codified.
- In table 2, patient #16, calorie intake: 14656?
- It seems that the insertion of Table 3 in the vicinity of the paragraph describing the table will be more legible and more convenient for the reader.
- Line 246: Reference to Table 4 but no table 4 in the manuscript. The table numbering should be ordered.
- Line 295: “group 60-“ Does it mean a group of patients under 60 years of age? The description of 50+ and 40+ seems more elegant, patients were divided into such groups
- I find the description of the groups in table 5 a bit unclear. For example, what is the difference between BN with ≥ 1NB and No BN with NB. Please explain clearly under the table.
Author Response
We would like to thank to the Reviewer 1 for her/his effort. In the attachment is cover letter, point by point response to the reviewers, English correction confirmation, manuscript with track changes and with accepted changes, and supplementary materials

Reviewer 2 Report
This manuscript describes 3-week long study of 26 obese inpatients on controlled 567 kcal diet administered in hospital setting. Beside loss of weight and other anthropometric parameters, standard blood tests at the beginning and the end of trial, genomic instability was assessed in lymphocytes by CBMN assay, GSH and ROS were measured in serum. The results indicate that this short-term intervention improved lipid profile urea and insulin resistance and decreased genomic instability. The variability of results suggests that an individual approach to obesity treatment is advisable and may require such specialized tests. Unfortunately, the paper is far from suitable for publication. There are many errors in style and grammar, too many to point out in a review. Paying more attention to any Word program corrections would eliminate some of them, but a professional editor familiar with writing scientific articles in English is probably required. The text is often difficult to read and understand. The sentences are too long, convoluted, repetitious, using too many words (not concise), and even contradictory. On other hand, the descriptions lack precision, the abbreviations are overused and some never explained (f.e. NER on line 475, NDI on line 488).
The very title of this paper belongs to Conclusions or the Abstract and is one of the longest ever. “The effects of 3-week hospital-controlled very low-calorie diet in severely obese patients” would probably be sufficient.
The quotations in the text do not correspond sometimes to the numbers in References. For example, EPIC-Norfolk questionnaire is referred to [21] on line 112. Reference 21 is about the comet assay. On line 579, reference 72 is quoted while it should be probably 71. All reference numbers in the whole text should be double checked. The authors frequently refer to their “previous study (Ozvald et al. 2021, under revision)”. Readers cannot find such article and it is not certain that it will be published. Such reference can be permitted only when both articles are published together in one issue of the same journal. Maybe this publication should be submitted after the ‘previous study’ is published online?
The long string of numbers, often without necessary spaces (f.e. lines 293-299, 306-309, 316-318) are awkward to read. The numbers after the decimal point should be rounded uniformly and placed in tables or illustrated by graphs. On lines 311 and 312, a comma is used instead of decimal point. When subjects were subdivided according to age, the classes were denoted as 60+, 60-, 50+, 40+ (lines 293-297) instead of >60 vs.<60, and more precise 51-60, 41-50 years old.
The standardized food questionnaire EPIC-Norfolk (line 112) should be explained in more detail. It is a food frequency questionnaire, and it can be made country specific. Was it adjusted to typical Croatian foods? The extreme under-reporting by many subjects (Table 2) may have been due to the absence of typical regional foods, especially mixed dishes, in the questionnaire. On other hand, subject 16 reported an impossibly high intake of 14656 kcal/day. The table does not specify CI units or symbol S (smoking). Did the smokers continue to smoke in the hospital for 3 weeks?
Why certain parameters in Table 5 are in bold script and why only some of them have p-value calculated (non-significant)? Serum cholesterol should be specified everywhere as total cholesterol (TC), LDL-cholesterol (LDL-C) or HDL-cholesterol (HDL-C). Triglycerides should be abbreviated as TG.
There are also typing mistakes, like splitting words in Abstract (glutathi-one, suscep-tible, use-fulness), or ‘alt-hough’ in lines 560-561. It should be ‘abasic sites’ on line 516, and ‘multiple inner organs’ on line 437.
There is no such a thing as “positive decrease” (Figure 2) or “double decrease” (line 282, 430).
The authors did not discuss possible adverse effects of such very low-calorie diet and seem to consider it acceptable for extended time (3 months?). However, insufficient intake of protein, calcium and many vitamins (Table 1) would result in a great loss of muscle mass (including heart muscle), bone density, and other problems, unless accompanied by an adequate supplementation regimen.
The manuscript needs a lot of work before it may be considered for publication. It is difficult to imagine that all seven authors have read this manuscript (it is not “the published version”, as stated on line 578) and nobody noticed the obvious errors.
Author Response
We would like to thank to the Reviewer 2 for his/her effort. In the attachment are cover letter with response to the Reviewers, English correction confirmation, manuscript with track and accepted changes and supplementary material

Round 2
Reviewer 2 Report
The revised manuscript is greatly improved in substance and style. The reference numbers were corrected, as well as tables, the title, and the individual results were added in Table 3. However, the revision is not easy to read because it contains all corrections in two different colors. It would be advisable to provide a clean revised copy that may reveal the remaining errors and problems. The sentences tend to be still too long and convoluted, sometimes they seem to lose the thread of the argument, especially in the Discussion. Below are some corrections and questions:
It is better not to split words at the end of a line, because the dash may remain in the text in the middle of line after corrections (see f. e. splits in lines 23,24, etc., and the unnecessary dashes in 27, 31 in mi-cronucleus and glutathi-one, or co-morbidities on line 463). The problem was pointed out before.
Line 28. It should be “26 obese patients”.
Line 45. It should mention the time period for 57 million deaths.
Line 46,50. Use different types of brackets.
Line 56. It is “high”, not “highly”.
Line 77. It should be “by measurement (or assays) of serum..., leukocyte…DNA…”
Line 143. It should be “daily meals freshly prepared…”
Line 150 and Table 1. It should be “protein”, not “proteins”.
Line 155. It should be “measured”, not “analyzed”.
Line 158. It should be “to calculate the percentage of excessive body weight loss”.
Line 162. It should be “as possibly affected…”
Line 177. Is it a 25 cm3 flask?
Line 222. ROS is plural, so it should be “were”.
Line 249. It should be “Their regular therapy, if any, was maintained during the study”. Remove “and for those… excluded from the study” in the next lines.
Line 257. Remove “or nearby”.
Table 2 title. It should be “tumour incidence”, not “tumours incidence” This grammatical error repeats in the article. If you insist on plural, the Genitive is “tumours’ incidence” or “Incidence of tumours”.
Line 286-292. It is not necessary to keep the last zero after the decimal point for the sake of uniformity unless significant (also lines 520-522, 565-566, 587-588).
Line 292. Remove “rate”.
Line 301. It should be “After the diet, the glucose levels were significantly lower and fell in the range of normal values.”
Table 4. It would be better to use an arrow (symbol) → instead of a long dash, which usually represents range.
Line 386. The study measures a permanent DNA damage, as described in many places, so it cannot be “repaired”.
Line 390. Remove “though”.
Line 392-393. It should be “…decrease by half after the diet, demonstrating...”
Line 402-406. It is a non sequitur: “while values for vitamin B12... after the diet.” It does not follow “equal number of females and males”. Use two separate sentences.
Line 447. Remove “to be able”.
Line 465. Where in the body was this increase of free fatty acids?
Line 478-479. Remove “changes of”, and “Even only”. It should be “The loss of excess body fat lowers…”
Line 494. Replace “even” with “preferably”.
Line 524. It should be “on… VLCD”, not “in”.
Line 527. Remove “even with the need of”.
Line 529. It should be (Mancini et al., 39) and “100 obese subjects”
Line 532. It is “programme“(British) for the sake of uniformity. “Program” is American English.
Line 533. Use “study” instead of “programme“.
Line 534. It should be “changes in medications were made for several participants…”
Line 544. It should be “obese subjects”. Line 545. Remove “still”.
Line 551-552. Period after BMR. Start a new sentence from “our diet…”
Line 556. Remove “with an unexpected decrease in HDL-C”.
Line 558. Remove “that”. Insert reference number [52] after “Lips et al.”
Line 560. It should be “The decrease was nearly 50% of the values before diet (or study).”
Line 565. Remove “group”. Remove zeroes (see line 286 above).
Line 569. It should be “inner organ DNA damage” (see Table 2 above).
Line 571. Insert reference number [42] after “our previous study”.
Line 577-579. It should be “Increased fT4 but not fT3… stopping conversion of elevated fT4 to fT3…”
Line 583. It should be “levels”. Line 584. It should be “in another study…” and “…on VLCD”.
Line 585. It should be “similar to the values…”
Line 590. It should be “not comparable”, and “3-week treatment”. Remove “with expected lower values than in our study”. Such statements (expected, unexpected) may imply research bias.
Line 597. It should be “after the study, so…”. Line 598. “this patient continues to control…”
Line 599. Remove “controlled”.
Line 600. Period after “checkups”. New sentence: ”This patient has achieved…”
Line 601. It should be “Most of the other patients’ medications…”.
Line 602, 603, 606. Use “study” instead of “diet”.
Line 610. Remove “even”. Line 611-612. Remove ”the gender mixed population and”.
Line 615. Replace “good” with “useful”. Insert “patients” after ”obese”.
Line 619. Insert reference number [42] after “VLDC”.
Line 623. Replace “The reason” with “It”.
Line 625. This sentence is not clear. There was a decrease in leukocyte count after the diet, not an increase, and the erythrocyte count was not reported.
Line 634. It should be “by Picklo”.
Line 643-644. It should be “these measurements”.
Line 648. Remove “in the body”.
Line 657. Comma after “people”.
Line 660. Replace an interventional study” with “intervention”.
Line 662. Remove “after 52 weeks or” A year has 52 weeks. It should be “slight”, not “slightly”.
Line 678. Insert reference number [65] after “et.al. and remove “from 2014“.
Line 681. Remove ”was only comparable and”.
Line 683. Replace “confirming with “confirms”. Line 684. Replace “as” with “play”.
Line 686. Period after [68,69]. Start a new sentence from “Long-term follow-up…”.
Line 688. Remove “adverse”.
Line 691. Screening for obesity does not require CBMN assay, anthropometric measurements are sufficient.
Line 696. It should be ”3-week-567 kcal-…” here and everywhere.
Line 709. It should be “advocated”.
Line 711. Is it “entered” or “finished”?
Line 718. It should be “55 g carbohydrates/day”.
Line741. It should be ”patient consent”.
Most of the corrections are minor, but their abundance detracts from the value of this study. The manuscript must be carefully edited before publication.

Author Response
Zagreb, 7th December 2021
Cover letter for revised submission
Dear Editors of the Nutrients Journal and Editors of the Special Issue Biomarkers of Nutritional Exposure and Nutritional Status,
Thank You for the evaluation of our manuscript nutrients-1477140 "Effects of a 3-week hospital-controlled very low-calorie diet in severely obese patients" which we as the authors (Ivan Ožvald, Dragan Božičević, Lidija Duh, Ivana Vinković Vrček, Ivan Pavičić, Ana-Marija Domijan and Mirta Milić) submitted to your Journal.
We highly appreciate the valuable comments of the Reviewers 2 regarding our manuscript. The suggestions and remarks were helpful, and we incorporated them into the revised paper.
Please find attached the revised manuscript but without Track Changes as requested by the Reviewer 2 (all the requirements are mentioned in the Reviewer´s response with also the lines in the manuscript where the changes have been made, since also the clean reviewers version was completely different than ours), and our Response to Reviewer´s 2 Comments, which provides answers to the Reviewer 2 and a detailed description of the changes made during the process of revision (point by point).
We sincerely hope that the Reviewer 2 and you, the Editors in Chief of the Special Issue, and the main Editors will be satisfied with the responses and revisions of the original manuscript.
We thank you for your efforts and look forward to hearing from you at your earliest convenience.
With my best regards,
Mirta Milić, PhD, Corresponding author
Mutagenesis Unit,
Institute for Medical Research and Occupational Health, Zagreb, Croatia
Response to Reviewer 2
The Authors would like to thank Reviewer #1 for their valuable comments on our manuscript. We have responded to all the comments and revised the paper accordingly. Detailed responses to each comment are provided below.
The revised manuscript is greatly improved in substance and style. The reference numbers were corrected, as well as tables, the title, and the individual results were added in Table 3. However, the revision is not easy to read because it contains all corrections in two different colors. It would be advisable to provide a clean revised copy that may reveal the remaining errors and problems. The sentences tend to be still too long and convoluted, sometimes they seem to lose the thread of the argument, especially in the Discussion. Below are some corrections and questions:
- It is better not to split words at the end of a line, because the dash may remain in the text in the middle of line after corrections (see f. e. splits in lines 23,24, etc., and the unnecessary dashes in 27, 31 in mi-cronucleus and glutathi-one, or co-morbidities on line 463). The problem was pointed out before.
Response to the Reviewer2: Thank You for Your remarks. As requested by the Journal, in the same manuscript there was a version with track changes and later the clean version with all accepted changes. One colour was for authors changes according to the requests of the Reviewers and the other colour was from changes made by the English official sworn court interpreter for the English language and scientific articles.
As for the words that are broken, it is the matter of the Nutrients Journal requested use of template for publishing the manuscripts, and we cannot change that. In addition, as we mentioned before, the reading version depends on each Word (Office) version in which the manuscript is opened. The lines you are mentioning are not the same when we open the manuscript on our computers, and that will be the part for Nutrients editing part to prepare it for final version for publishing.
- Line 28. It should be “26 obese patients”.
Response to the Reviewer 2: In our document, that is a line 30, and we have changed the sentence according to your requests into:
A correlation was found between GSH decrease and reduced DNA damage. VLCD revealed susceptible individuals with remaining higher DNA damage for further monitoring. In a highly heterogeneous group consisting of 26 obese patients with class II and III obesity, the approach demonstrated its usefulness and benefits in health improvement, enabling an individual approach to further monitoring, diagnosis, treatment and risk assessment based on both anthropometric/biochemical VLCD changing parameters and CBMN results.
- Line 45. It should mention the time period for 57 million deaths. Line 46,50. Use different types of brackets.
Response to the Reviewer 2: Thank You for Your comment, but we need to say that the time-period has already been mentioned in the text. You can find it in the line 41. We have also changed the type of brackets, but that again at the end will depend on the Nutrition’s Journal final editing. The whole sentence is:
Non-communicable diseases (NCDs), characterized as non-transmissible with a long duration and slow progression, account for 63 % of total deaths worldwide {up to 36 million of 57 million total deaths according to the World Health Organization´s (WHO) Global Action Plan for the prevention and control of NCDs 2013–2020} [1].
- Line 56. It is “high”, not “highly”.
Response to the Reviewer 2: Thank You for Your comment, but after consulting 4 more English speakers, we do think it is highly. In our manuscript the line is 61. The sentence is:
The programme has been in place for five years and has proven to be a highly successful and low cost programme that not only improves patient wellbeing, but also offers a strong opportunity to reduce the burden on the health care system associated with the incidence of severe chronic disease and comorbidities in obese people, since a healthy diet is considered both primary and secondary prevention of NCD (further) development.
- Line 77. It should be “by measurement (or assays) of serum..., leukocyte…DNA…”
Response to the Reviewer 2: Thank You for Your remarks, but in the whole paragraph around line 77, or line 82 (as we think that is the line difference from Your version and ours, according to those 4 reviewers requests answered already), in paragraphs before and after that, we do not find anything that would respond to your request for changing.
- Line 143. It should be “daily meals freshly prepared…”
Response to the Reviewer 2: Thank You for Your remark, in line 122 we have changed the order of the word correspondingly into: three daily meals freshly prepared
- Line 150 and Table 1. It should be “protein”, not “proteins”.
Response to the Reviewer 2: Thank You for Your remark, in our case is line 129 and Table 1, but since all the words are in plural, we would like to leave also proteins in plural
- Line 155. It should be “measured”, not “analyzed”.
Response to the Reviewer 2: Thank You for Your remark, in our case is line 134, we have changed the word analyzed with measured
- Line 158. It should be “to calculate the percentage of excessive body weight loss”.
Response to the Reviewer 2: Thank You for Your remark, in our case are line 137 and line 138. We have changed that part of the sentence as requested.
- Line 162. It should be “as possibly affected…”
Response to the Reviewer 2: Thank You for Your remark, in our case is line 141. We have changed that part of the sentence as requested.
- Line 177. Is it a 25 cm3flask?
Response to the Reviewer 2: Thank You for Your remark, but for the flasks in which the cells are grown, the most important thing is the surface area, although our blood samples does not have adherent cells in normal growing conditions, but that is the way in which you describe the flask for cell growing and for ordering them.
- Line 222. ROS is plural, so it should be “were”.
Response to the Reviewer 2: Thank You for Your remark, in our case is line 198. We have changed that as requested.
- Line 249. It should be “Their regular therapy, if any, was maintained during the study”. Remove “and for those… excluded from the study” in the next lines.
Response to the Reviewer 2: Thank You for Your remark, in our case is line 224. We decided not to change the sentences, since almost all the patients (except 2) did have therapy.
- Line 257. Remove “or nearby”.
Response to the Reviewer 2: Thank You for Your remark, in our case is line 231. We decided not to change the sentence, since in Croatian laws specifically is mentioned inside the building and 20-50 meters from the building (nearby) (depending on the type of the hospital,etc…)
- Table 2 title. It should be “tumour incidence”, not “tumours incidence” This grammatical error repeatsin the article. If you insist on plural, the Genitive is “tumours’ incidence” or “Incidence of tumours”.
Response to the Reviewer 2: Thank You for Your remark. Beside Table 2, we have found only one word in the manuscript that needed a change, and have changed it accordingly.
- Line 286-292. It is not necessary to keep the last zero after the decimal point for the sake of uniformity unless significant (also lines 520-522, 565-566, 587-588).
Response to the Reviewer 2: Thank You for Your remark. In Your first review, you have asked for all the decimal numbers to have the same number of decimal number, so we have changed everything accordingly, except p values, and that explanation about uniform decimal numbers is written in the statistical analysis part, so we have decided to leave it as it was requested for the first time.
- Line 292. Remove “rate”.
Response to the Reviewer 2: Thank You for Your remark, in line 259, we have removed the word rate.
- Line 301. It should be “After the diet, the glucose levels were significantly lower and fell in the range of normal values.”
Response to the Reviewer 2: Thank You for Your remark, in lines 266 and 267; we have changed the sentence accordingly.
- Table 4. It would be better to use an arrow (symbol) → instead of a long dash, which usually represents range.
Response to the Reviewer 2: Thank You for Your remark; we have changed the long dash with an arrow symbol in the Table 4.
- Line 386. The study measures a permanent DNA damage, as described in many places, so it cannot be “repaired”.
Response to the Reviewer 2: Thank You for Your remark; we have removed repaired from the sentence in the line 282.
- Line 390. Remove “though”.
Response to the Reviewer 2: Thank You for Your remark; we have removed the word though, line 285.
- Line 392-393. It should be “…decrease by half after the diet, demonstrating...”
Response to the Reviewer 2: Thank You for Your remark; we have changed that part of the sentence, line 287 and 288
- Line 402-406. It is a non sequitur: “while values for vitamin B12... after the diet.” It does not follow “equal number of females and males”. Use two separate sentences.
Response to the Reviewer 2: Thank You for Your remark; we have changed thhe sentence into two sentences, line 295-298
As the group contained only two smokers, the smoking factor could not be analysed. The group consisted of the same number of females and males. The values for vitamin B12 and folic acid (measured to check for possible vitamin and mineral deficiency as an exclusion factor, results not shown) did not differ before and after the diet.
- Line 447. Remove “to be able”.
Response to the Reviewer 2: Thank You for Your remark; we have removed to be able from the sentence, line 333,334
- Line 465. Where in the body was this increase of free fatty acids?
- Response to the Reviewer 2: Thank You for Your remark; we have added words from adipose tissue, as demonstrated by the works that we are citing, and now in the lines 346-348 is the sentence:
Pathophysiological mechanisms underlying obesity are explained with the increase of free fatty acids released from adipose tissue, lipid intermediates, insulin resistance with excess total and intra-abdominal adipose tissue, and inflammation [32–37].
- Line 478-479. Remove “changes of”, and “Even only”. It should be “The loss of excess body fat lowers…”
Response to the Reviewer 2: Thank You for Your remark; the phrase changes of does not exist in the manuscript, and we would not want to remove the phrase Even only, because usually there are more factors that are influencing the levels of cancer risk and here is implemented that even only one small change in one factor can have really significant potency in cancer risk decrease. Line 360
- Line 494. Replace “even” with “preferably”.line 394
Response to the Reviewer 2: Thank You for Your remark; the word even was replaced as requested, line 394.
- Line 524. It should be “on… VLCD”, not “in”.
Response to the Reviewer 2: Thank You for Your remark; but we are referring to the results in studies dealing with VLCD………demonstrated in VLCD studies… line 406
- Line 527. Remove “even with the need of”.
Response to the Reviewer 2: Thank You for Your remark; but not all the patients in the study changed the therapy afterwards, some of them, and that is also something important to point out with that phrase that you want us to delete. line 403
- Line 529. It should be (Mancini et al., 39) and “100 obese subjects”
Response to the Reviewer 2: Thank You for Your remark, as for the reference we are again referring to the Nutrients Journal referencing rules, and as for the word subjects, we have added that word as you requested, although obese is a term regularly in use.
- Line 532. It is “programme“(British) for the sake of uniformity. “Program” is American English.
Response to the Reviewer 2: Thank You for Your remark, we have changed the word into programme, line 411
- Line 533. Use “study” instead of “programme“.
Response to the Reviewer 2: Thank You for Your remark, we have changed the word programme with the word study, line 412
- Line 534. It should be “changes in medications were made for several participants…”
Response to the Reviewer 2: Thank You for Your remark, we have removed the phrase for several participants, and put it on the place in the sentence you requested, lines 412 and 413
- Line 544. It should be “obese subjects”.
Response to the Reviewer 2: Thank You for Your remark, we have removed the word patients into subjects as requested, line 415
- Line 545. Remove “still”.
Response to the Reviewer 2: Thank You for Your remark, we have removed the word still as requested, line 422
- Line 551-552. Period after BMR. Start a new sentence from “our diet…”
Response to the Reviewer 2: Thank You for Your remark, we have put the dot after BMR and started new sentence with Our diet…, as requested, lines 427-430
- Line 556. Remove “with an unexpected decrease in HDL-C”.
Response to the Reviewer 2: Thank You for Your remark, we have removed that part of the sentence as requested, line 431
- Line 558. Remove “that”. Insert reference number [52] after “Lips et al.”
Response to the Reviewer 2: Thank you for this observation, we have removed the word that. We did not notice that in this case we did not respect the rules of the Journal of putting the references at the end of the sentence. We have corrected it according to the Journals rules, lines 434-440
- Line 560. It should be “The decrease was nearly 50% of the values before diet (or study).”
Response to the Reviewer 2: Thank you for this observation, we have changed the sentence accordingly, ines 435,436
- Line 565. Remove “group”. Remove zeroes (see line 286 above).
Response to the Reviewer 2: Thank you for this observation, we have removed the word group, line 438. As requested in the first revision, all the numbers with decimal places were made uniform according to the number of decimal places (2), except for p values. So we will stay with the uniformity of the numbers.
- Line 569. It should be “inner organ DNA damage” (see Table 2 above).
Response to the Reviewer 2: Thank you for this observation, but this phrase was the request in the first round of revisions, so it will stay like that as requested before.
- Line 571. Insert reference number [42] after “our previous study”.
Response to the Reviewer 2: Thank You for Your remark, we have added the reference number 42 according to the rules of Journal reference style, line 446
- Line 577-579. It should be “Increased fT4 but not fT3… stopping conversion of elevated fT4 to fT3…”
Response to the Reviewer 2: Thank You for Your remark, we have changed the sentences accordingly, line 450, line 451
- Line 583. It should be “levels”.
Response to the Reviewer 2: Thank You for Your remark, we have changed it accordingly, line 455
- Line 584. It should be “in another study…” and “…on VLCD”.
Response to the Reviewer 2: Thank You for Your remark, we have changed it accordingly, line 456
- Line 585. It should be “similar to the values…”
Response to the Reviewer 2: Thank You for Your remark, we have changed it accordingly, line 457
- Line 590. It should be “not comparable”, and “3-week treatment”. Remove “with expected lower values than in our study”. Such statements (expected, unexpected) may imply research bias.
Response to the Reviewer 2: Thank You for Your remark, we have changed not-comparable and 3-week treatment, and removed expected but we will leave the rest of the text, as it is important also to say that their values were lower than ours (3-week treatment and their 3-month treatment with lower values than in our study), lines 461, 462
- Line 597. It should be “after the study, so…”.
Response to the Reviewer 2: Thank You for Your remark, we have change it accordingly, line 472
- Line 598. “this patient continues to control…”
Response to the Reviewer 2: Thank You for Your remark, we have change it accordingly, line 468
- Line 599. Remove “controlled”.
Response to the Reviewer 2: Thank You for Your remark, we have removed the word, line 469
- Line 600. Period after “checkups”. New sentence: ”This patient has achieved…”
Response to the Reviewer 2: Thank You for Your remark, we have changed the sentences accordingly, line 470
- Line 601. It should be “Most of the other patients’ medications…”.
Response to the Reviewer 2: Thank You for Your remark, we have made the changes accordingly, line 470, line 471
- Line 602, 603, 606. Use “study” instead of “diet”.
Response to the Reviewer 2: Thank You for Your remark, we have made the changes accordingly, line 471
- Line 610. Remove “even”.
Response to the Reviewer 2: Thank You for Your remark, we have made the changes accordingly, line 479
- Line 611-612. Remove ”the gender mixed population and”.
Response to the Reviewer 2: Thank You for Your remark, we have made the changes accordingly, line 480
- Line 615. Replace “good” with “useful”. Insert “patients” after ”obese”.
Response to the Reviewer 2: Thank You for Your remark, we have made the changes accordingly, lines 483,483
- Line 619. Insert reference number [42] after “VLDC”.
Response to the Reviewer 2: Thank You for Your remark, we have made the changes accordingly to the Journal requirements, line 489
- Line 623. Replace “The reason” with “It”.
Response to the Reviewer 2: Thank You for Your remark, we have made the changes accordingly, line 490
- Line 625. This sentence is not clear. There was a decrease in leukocyte count after the diet, not an increase, and the erythrocyte count was not reported.
Response to the Reviewer 2: Thank You for Your remark, we have made the changes and have put explanation after erythrocytes: (leukocytes, erythrocytes-results not shown in this study), line 492, line 493
- Line 634. It should be “by Picklo”.
Response to the Reviewer 2: Thank You for Your remark, we have made the changes accordingly, line 501
- Line 643-644. It should be “these measurements”.
Response to the Reviewer 2: Thank You for Your remark, we have made the changes accordingly, line 510
- Line 648. Remove “in the body”.
Response to the Reviewer 2: Thank You for Your remark, we have made the changes accordingly, line 514
- Line 657. Comma after “people”.
Response to the Reviewer 2: Thank You for Your remark, we have made the changes, line 522
- Line 660. Replace an interventional study” with “intervention”.
Response to the Reviewer 2: Thank You for Your remark, we have made the changes accordingly, line 525
- Line 662. Remove “after 52 weeks or” A year has 52 weeks. It should be “slight”, not “slightly”.
Response to the Reviewer 2: Thank You for Your remark, we have made the changes accordingly, line 526, line 527
- Line 678. Insert reference number [65] after “et.al. and remove “from 2014“.
Response to the Reviewer 2: Thank You for Your remark, we have made the changes according to the Journals requirements, lines 541-544
- Line 681. Remove ”was only comparable and”.
Response to the Reviewer 2: Thank You for Your remark, we have made the changes accordingly, line 544
- Line 683. Replace “confirming with “confirms”.
Response to the Reviewer 2: Thank You for Your remark, it writes actually confirm and it should stay like that, there are more evidence than one (plural), line 545
- Line 684. Replace “as” with “play”.
Response to the Reviewer 2: Thank You for Your remark, we have made the changes accordingly, line 545
- Line 686. Period after [68,69]. Start a new sentence from “Long-term follow-up…”.
Response to the Reviewer 2: Thank You for Your remark, we have made the changes accordingly, line 548, 549
- Line 688. Remove “adverse”.
Response to the Reviewer 2: Thank You for Your remark, we have made the changes accordingly, line 550
- Line 691. Screening for obesity does not require CBMN assay, anthropometric measurements are sufficient.
Response to the Reviewer 2: Thank You for Your remark. Our group has enough published manuscripts with both CBMN and comet assay methods that demonstrated to be sensitive techniques for discovering individuals with higher cancer and mortality risk, even when the other diagnosis are still not visible, or disease are not developed. The assay is not meant to be diagnostic marker alone for obesity
- Line 696. It should be ”3-week-567 kcal-…” here and everywhere.
Response to the Reviewer 2: Thank You for Your remark. It was the English correction from the professional as requested by the reviewer and she changed it. We have change it again through the text, thank you…lines58,84,122,237,Table 3, Table 4, Table 5, line 290, 340,378,386,529,557.
- Line 709. It should be “advocated”.
Response to the Reviewer 2: Thank You for Your remark, we have made the changes accordingly, line 566
- Line 711. Is it “entered” or “finished”?
Response to the Reviewer 2: Thank You for Your remark. It is entered, the plan is to monitor further all the patients who entered in the programme, the programme lasts until they are not obese any more.
- Line 718. It should be “55 g carbohydrates/day”.
Response to the Reviewer 2: Thank You for Your remark, we have made the changes accordingly. Since all the measurements are without /, we have put per day. Line 576
- It should be ”patient consent”.
Response to the Reviewer 2: Thank You for Your remark, we have made the changes accordingly, line 594
The revised manuscript is in the attachment
